# An analysis toolbox to explore mesenchymal migration heterogeneity reveals adaptive switching between distinct modes

Hamdah Shafqat-Abbasi, Jacob M Kowalewski, Alexa Kiss, Xiaowei Gong, Pablo Hernandez-Varas, Ulrich Berge, Mehrdad Jafari-Mamaghani, John G Lock*[†], Staffan Strömblad[†]

Department of Biosciences and Nutrition, Karolinska Institutet, Huddinge, Sweden

**Abstract** Mesenchymal (lamellipodial) migration is heterogeneous, although whether this reflects progressive variability or discrete, 'switchable' migration modalities, remains unclear. We present an analytical toolbox, based on quantitative single-cell imaging data, to interrogate this heterogeneity. Integrating supervised behavioral classification with multivariate analyses of cell motion, membrane dynamics, cell-matrix adhesion status and F-actin organization, this toolbox here enables the detection and characterization of two quantitatively distinct mesenchymal migration modes, termed 'Continuous' and 'Discontinuous'. Quantitative mode comparisons reveal differences in cell motion, spatiotemporal coordination of membrane protrusion/retraction, and how cells within each mode reorganize with changed cell speed. These modes thus represent distinctive migratory strategies. Additional analyses illuminate the macromolecular- and cellular-scale effects of molecular targeting (fibronectin, talin, ROCK), including 'adaptive switching' between Continuous (favored at high adhesion/full contraction) and Discontinuous (low adhesion/inhibited contraction) modes. Overall, this analytical toolbox now facilitates the exploration of both spontaneous and adaptive heterogeneity in mesenchymal migration.

*For correspondence: john.lock@ki.se

[†]These authors contributed equally to this work

Competing interests: The authors declare that no competing interests exist.

## Introduction

Cell migration is a profoundly heterogeneous phenomenon. Indeed, cells can adopt several substantially different migration modalities, including multicellular, amoeboid, and mesenchymal (also termed lamellipodial or lamellipodial-driven) migration, which can all be utilized by a broad range of cell types, as well as lobopodial migration, which has been observed specifically in fibroblasts (*Friedl and Wolf, 2010*; *Sahai, 2005*; *Petrie and Yamada, 2015*; *Petrie et al., 2014*; *Welch, 2015*; *Friedl and Alexander, 2011*). These migration modes represent 'prespecified' cellular configurations (i.e. cell states) that are favored under particular conditions (*Friedl, 2004*). Switch-like conversion between these distinct modes is therefore part of the plastic, adaptive/compensatory response of cells to either environmental modulation (*Liu et al., 2015*; *Starke et al., 2014*; *Ruprecht et al., 2015*) or molecular targeting (*Sahai et al., 2007*; *Sanz-Moreno et al., 2008*; *Somlyo et al., 2003*; *Wolf, 2003*). At a finer scale, heterogeneity is also evident within these migration modes, arising either stochastically or as an adaptive response to changing cues (*Geiger et al., 2009*; *Lämmermann and Sixt, 2009*; *Lock et al., 2014*; *Winograd-Katz et al., 2009*). Yet, partly due to a lack of adequate quantification, it remains unclear to what extent variation within modes occurs either progressively along a continuum or in a switch-like manner between as yet undefined intra-modal subpopulations. Specifically, in the case of amoeboid migration, three discrete sub-modalities

**eLife digest** During an animal's lifetime, many of its cells will move from one location in the body to another. For example, skin cells can migrate to repair wounds. Prior to migration, cells are usually attached to a scaffold called the extracellular-matrix, which helps to hold them in a particular location within a tissue. Individual cells can move in different ways. During a type of movement called mesenchymal migration, the front end of a cell grows outwards and attaches to a different section of the matrix. The rear of the cell is pulled forward and it detaches from the matrix and retracts, which allows the entire cell to move forward. In contrast, during amoeboid migration, the moving cells are only loosely attached to the matrix and move by gliding.

There are large variations in how cells move and they can adopt modes that lie between the two extremes of mesenchymal and amoeboid migration. They can also switch between modes depending on their requirements. Shafqat-Abbasi et al. developed a method to analyse how individual human lung cancer cells move. The method uses software to collect data on cell shape, speed of movement and other features from microscopy images of the migrating cells.

The experiments reveal that the cells adopt two distinct migration modes, which Shafqat-Abbasi et al. termed 'Discontinuous' and 'Continuous'. The majority of cells migrated in the Discontinuous mode, in which cells moved in many different directions. This was caused by a lack of coordination between the outgrowth of the front end of the cell, and the retraction of the back from the matrix. In contrast, in the cells that migrated using the Continuous mode, an outgrowth consistently led to a retraction, which enabled cells to move in one direction.

Further experiments revealed that the mode of migration used by the cells is affected by how tightly they are bound to the extracellular-matrix, and the mechanical forces generated inside the cells to drive the movement. Shafqat-Abbasi et al.'s method provides an analytical toolbox that other researchers can use to study the mesenchymal migration of animal cells.

have been observed, and these can co-exist under individual conditions (*Welch, 2015*; *Lämmermann and Sixt, 2009*; *Yoshida, 2006*). By contrast, potentially distinct styles of mesenchymal (lamellipodial) migration, including keratocyte-like (*Barnhart et al., 2015*; *Keren et al., 2008*) and fibroblast-like (*Abercrombie et al., 1977*; *Theisen et al., 2012*) migration, have been described as arising largely in separate cell types or conditions. Therefore, despite some early suggestions (*Lewis et al., 1982*), it has remained uncertain to what extent divergent sub-modalities of mesenchymal migration spontaneously emerge in parallel within uniform cell populations and conditions, and whether these modes are truly quantitatively distinct, or instead represent extremes in a broad phenotypic continuum. These questions are important because each discrete migration mode, with its unique internal logic and dependencies, may respond differently to altered regulatory cues – thereby defining a multifaceted adaptive response. Such responses are exemplified by the seminal observation that amoeboid motility provides a compensatory escape route for tumor cells when mesenchymal migration is blocked via inhibition of proteolytic mechanisms or by confinement (*Wolf, 2003*). Identifying and characterizing such distinct modes within individual cell types and/or conditions may therefore be key to understanding heterogeneity in response to experimental or even clinical interventions.

Unlike amoeboid migration, which requires little or no cell-matrix adhesion (*Lämmermann et al., 2008*), mesenchymal migration demands cell adhesion to the extracellular matrix (ECM) for effective force application (*Friedl and Wolf, 2010*). This adhesion is achieved through the interaction of integrin-mediated cell-matrix adhesion complexes (CMACs) with ECM ligands, while applied forces are generated by the F-actin cytoskeleton and actomyosin system (*Geiger et al., 2009*; *Humphries et al., 2015*). This arrangement defines the ECM – adhesion – F-actin axis that directly mediates mesenchymal migration (*Parsons et al., 2010*; *Schwartz, 2010*).

Integrins are central to the ECM – adhesion – F-actin axis and are both anchored and activated by binding through their extracellular domains to ECM ligands, such as collagen, laminin, and fibronectin (*Lewis et al., 1982*). The composition, topology, and density of ECM ligands therefore play key roles in mesenchymal migration (*Geiger et al., 2009*). Through interaction with integrin tails,

talin acts a critical intracellular regulator of integrin activation (*Calderwood et al., 1999*; *Kiss et al., 2015*; *Moser et al., 2009*; *Tadokoro, 2003*), and is also one of several CMAC proteins linking integrins to the F-actin cytoskeleton (*Schwarz and Gardel, 2012*). This enables cellular force application from F-actin, through CMACs, to the ECM, thereby driving membrane protrusion/retraction and cell translocation (*Small and Resch, 2005*). Furthermore, CMACs function both up- and downstream of the small GTPases Rac1 and RhoA (*Raftopoulou and Hall, 2004*), which modulate both F-actin polymerization and actomyosin contractility, the latter through the Rho kinase (ROCK)-mediated regulation of non-muscle myosin II (*Ridley, 2003*; *Riento and Ridley, 2003*). In fact, CMACs function as hubs for bi-directional chemical and mechanical information transduction across the plasma membrane, providing command and control of mesenchymal migration (*Hynes, 2002*; *Lock et al., 2008*).

Given the pivotal roles of CMACs and F-actin, their assessment can efficiently provide a broad estimation of how the complex machinery underlying mesenchymal migration is organized (*Lock et al., 2014*; *Gardel et al., 2010*; *Gupton and Waterman-Storer, 2006*; *Kim and Wirtz, 2013*). Such an approach, generally based on quantitative imaging, is especially effective when multiscale data capturing both cell behavior (migration) and organization (e.g. CMAC and F-actin status) is derived simultaneously on a per cell basis. This facilitates the leveraging of natural or induced heterogeneity to define: i) the statistical structure of variation (e.g. progressive or discrete) within and between cell populations, as well as; ii) key trends, dependencies and relationships within the cell migration system (*Lock et al., 2014*; *Keren et al., 2008*; *Kiss et al., 2015*; *Kowalewski et al., 2015*; *Ku et al., 2012*; *Lee et al., 2015*; *Lock and Strömblad, 2010*). Here, we extend on this approach by employing a tailored analytical toolbox to detect two quantitatively distinct sub-modalities of mesenchymal migration, thus illuminating the discrete nature of variation within this general migration archetype. We employ this unique suite of analytical tools to further characterize key aspects of the behavior, organization, and regulation of these divergent migration strategies. Collectively, this study now provides conceptual and practical capabilities to the cell migration research community, while also highlighting the importance of distinguishing individual mesenchymal migration modes as a vital precursor to understanding mesenchymal migration as a whole.

## Results

### The cell adhesion and migration analysis toolbox

To enable the detection of discrete mesenchymal migration sub-modalities, and thereafter, the comprehensive analysis of cellular organization, regulation, and adaptation underlying each mode, we have integrated a unique combination of analytical tools. These are now made freely available as a Matlab Toolbox: 'The Cell Adhesion and Migration Analysis Toolbox' – along with the raw quantitative data underlying this study, sample image data to aid implementation, and explanatory documentation (see doi:10.5061/dryad.9jh6m). The features included in this Toolbox are also specified briefly in 'Materials and methods'.

### Classification of 'Discontinuous' and 'Continuous' modes of mesenchymal migration

H1299 (non-small cell lung carcinoma) cells stably expressing EGFP-paxillin (marker for CMACs) and RubyRed-LifeAct (marker for F-actin) (H1299 P/L cells [*Lock et al., 2014*]) were imaged via confocal microscopy on glass coated with 2.5 µg/ml fibronectin (FN). H1299 P/L cells moved individually and exclusively via mesenchymal (lamellipodial) migration. Nonetheless, we observed and classified two qualitatively different migration modes emerging within the clonally derived H1299 P/L cell population. We termed these migration modes 'Discontinuous' and 'Continuous', reflecting their contrasting stepwise and smooth motion, respectively (*Figure 1*). Specifically, cells migrating in the Discontinuous mode cycle in sequence through at least three recognizable stages of movement, which we term 'lateral protrusion', 'polarization', and 'tail retraction', as described in *Figure 1A* (also see *Video 1*). These stages produce dramatic changes in cell morphology and frequent changes in migratory direction. Such directional changes occur at least in part because lateral protrusions typically develop at approximately 90° to the axis of the preceding tail retraction. Overall, the stepwise nature of Discontinuous migration is highly reminiscent of previous descriptions of fibroblast-like migration (*Theisen et al., 2012*; *Abercrombie et al., 1970*; *Chen, 1981*). By contrast, cells migrating

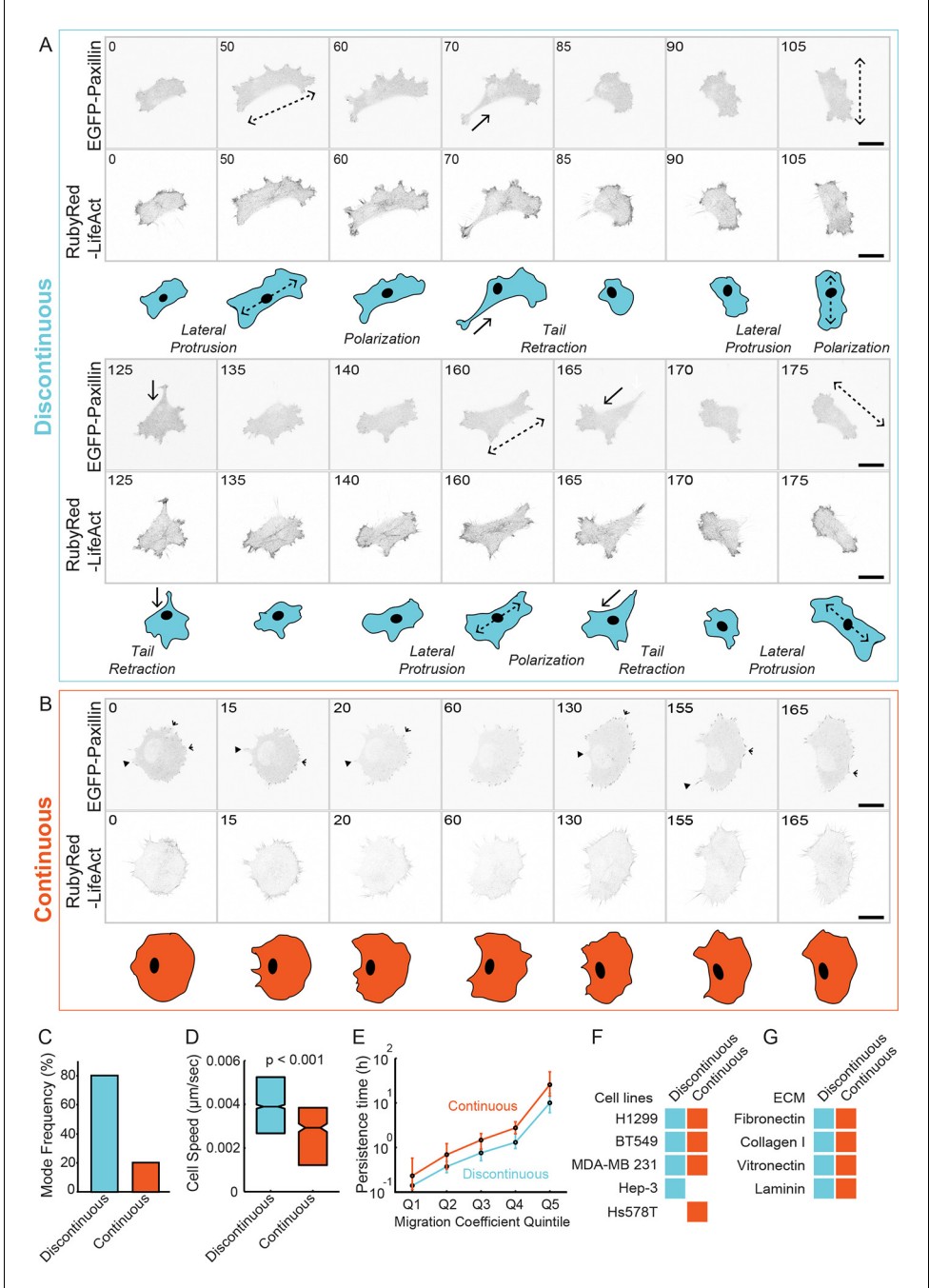

**Figure 1.** Classification of 'Discontinuous' and 'Continuous' modes of mesenchymal (lamellipodial) migration. (**A, B**) Confocal image sequences of individual H1299 cells stably expressing EGFP-paxillin (cell-matrix adhesion complex [CMAC] marker; upper image panels) and RubyRed-LifeAct (F-actin marker; lower image panels) migrating in either Discontinuous (**A**) or Continuous (**B**) modalities on fibronectin (FN)-coated (2.5 µg/ml) glass. Images are displayed with gray-scale inverted. Numbers denote time (min). Schematics in blue (Discontinuous, **A**) and orange (Continuous, **B**) depict typical cell morphology changes associated with each migration mode. Note stepwise cycles of: *lateral protrusion* (in directions of dashed arrows); cell *polarization*; and *tail retraction* (in direction of solid arrows) that recur during Discontinuous migration, and that lateral protrusion tends to occur at 90° to the preceding tail retraction. During Continuous migration, cell morphology is relatively stable, with many small protrusion (open arrowheads) and retraction (closed arrowheads) events producing smooth movement. Movies corresponding to cells shown in A and B are available in supporting material (**Videos 1**,**2**). (**C**) Quantification of frequencies of Discontinuous and Continuous modes. (**D**) Box plots of cell speed (µm/sec) per migration mode. Boxplots show median values and inter-quartile ranges (IQR, 25% to 75%). Notches indicate median +/− 1.57 * IQR/$\sqrt{n}$ (approximates 95% confidence interval of the median, n = number of cell observations, see 'Materials and methods'). Statistical discernibility assessed by Wilcoxon rank sum test, p < 0.001. (**E**) Cell trajectories in each mode were assessed via mean squared displacement (MSD) analysis and divided into quintiles (20%

*Figure 1 continued on next page*

*Figure 1 continued*

bins) according to their migration cefficient (related to speed of movement). Median values +/−1.57 * IQR/√n (n = number of cell observations) of a second MSD measure, persistence time (related to migration direction stability), are plotted per migration coefficient quintile. Number of observations per quintile: Discontinuous = ~ 402; Continuous = ~ 95. (F) A table summarizes results from visual inspection of several cell lines migrating on 2.5 µg/ml FN, confirming the emergence of either one or both Discontinuous and Continuous migration modes in these cell types. Example image sequences are presented in *Figure 1—figure supplement 3*. (G) A table summarizes results from visual inspection of H1299 cells migrating on several different extracellular matrix ligands, confirming the emergence of both migration modes under each condition. Example image sequences are presented in *Figure 1—figure supplement 4*. Scale bars = 20 µm.

The following figure supplements are available for figure 1:

**Figure supplement 1.** Discontinuous and Continuous migration modes emerge in parallel under uniform conditions.

**Figure supplement 2.** Discontinuous and Continuous migration modes can spontaneously inter-convert (mode switching).

**Figure supplement 3.** Discontinuous and/or Continuous migration modes recur in multiple cell types.

**Figure supplement 4.** Discontinuous and Continuous migration modes recur in H1299 cells adhering to multiple extracellular matrix ligands.

---

in the Continuous mode move progressively, with less frequent changes in cell morphology and motile direction, in a manner analogous to classical keratocyte-like migration (*Figure 1B* and *Video 2*) (*Keren et al., 2008*). Importantly, these migration modes emerge spontaneously in parallel under uniform experimental conditions within a clonally derived cell line (*Figure 1—figure supplement 1* and *Video 3*). It is noteworthy that cells may traverse the same underlying substrate regions while stably occupying distinct modes (see *Video 3*), indicating that local environmental inconsistencies are not the principal cause of differential mode identity. Yet, inter-modal conversion is possible, occurring bi-directionally between modes (*Figure 1—figure supplement 2* and *Videos 4* and *5*), indicating their profound plasticity.

Under the conditions described above, ~80% of observations showed cells migrating via the Discontinuous mode, with the remaining 20% moving in the Continuous mode (*Figure 1C*, see 'Materials and methods' for sample numbers relating to all experimental data). Automated cell tracking (see 'Materials and methods', [*Lock et al., 2014*]) revealed that cells in the Discontinuous mode migrate significantly faster than during Continuous migration (*Figure 1D*). By applying mean squared

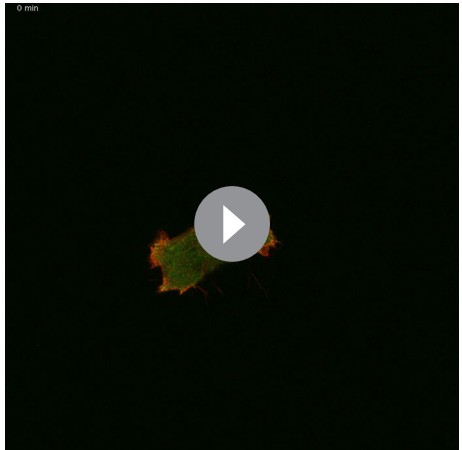

**Video 1.** H1299 P/L cell migration in the Discontinuous mode. High-resolution multiscale imaging of a single H1299 P/L cell expressing EGFP-paxillin (green, CMAC marker) and RubyRed-LifeAct (red, F-actin marker) during migration in the Discontinuous mode on 2.5 µg/ml fibronectin. Time in minutes shown.

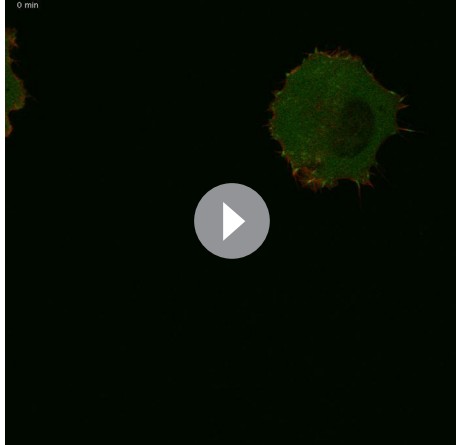

**Video 2.** H1299 P/L cell migration in the Continuous mode. High-resolution multiscale imaging of a single H1299 P/L cell expressing EGFP-paxillin (green, CMAC marker) and RubyRed-LifeAct (red, F-actin marker) during migration in the Continuous mode on 2.5 µg/ml fibronectin. Time in minutes shown.

displacement analysis of cell trajectories to assess measures related to cell speed (migration coefficient) and directionality (persistence time), we found that Discontinuous migration is less directionally stable at any given speed (*Figure 1E*). However, both modes of migration show a positive correspondence between speed and directionality, as recently reported (*Maiuri et al., 2015*). Importantly, an extensive comparison of migratory behaviors in several additional cell lines (*Figure 1F*, *Figure 1—figure supplement 3* and *Videos 6–11*) and during H1299 cell migration on alternative extracellular matrix ligands (*Figure 1G*, *Figure 1—figure supplement 4* and *Videos 12–17*) indicated that Discontinuous and Continuous migration modes are consistently recurring phenomena.

## Discontinuous migration is characterized by dramatic membrane retraction events

Cell migration is the product of membrane dynamics that can be divided into protrusive and retractive processes (*Ridley, 2003*). We quantitatively compared the spatial and temporal dynamics of membrane protrusions and retractions between and within each migration mode based on defining membrane dynamics over the minimal imaging interval of 5 min (*Figure 2*) (*Kowalewski et al., 2015*). This revealed that protrusions share similar size (area) distributions in both modes (*Figure 2A*), while retraction events are more extreme in size during Discontinuous migration, that is, more frequently very small or very large (*Figure 2B*). When compared within each mode, membrane retractions have a broader size distribution than protrusions in the Continuous mode (*Figure 2C*), although this is far more striking in the Discontinuous mode (*Figure 2D*). To investigate membrane dynamics operating over longer time-scales, we calculated the probabilities of protrusions and retractions of a given size based on net cell shape/position changes occurring over intervals of between 1 and 15 image frames (5 to 75 min). This revealed a relatively unstructured pattern of dynamics in Continuous cells, although retractions tended to be smaller and protrusions larger (*Figure 2E*). In contrast, Discontinuous migration consistently displayed a wide distribution of retraction sizes (very small and very large), with protrusion sizes uniformly moderate (*Figure 2F*). Notably, the relatively high probability of very large retraction events in Discontinuous cells corresponds with their observed tendency to undergo dramatic tail retraction events. Collectively, these results explain the spatial characteristics underlying stepwise or smooth cell motion during migration in Discontinuous or Continuous modes, respectively.

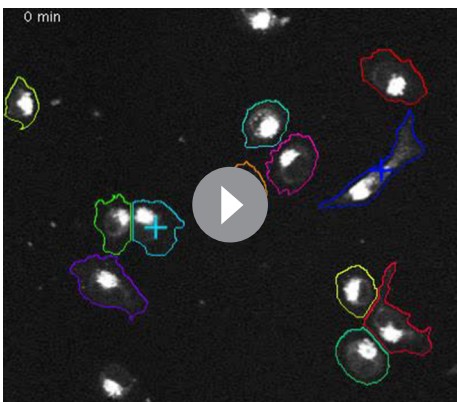

**Video 3.** Parallel emergence of Discontinuous and Continuous modes in H1299 P/L cells. Low-resolution imaging of H1299 P/L cells labeled with a membrane dye shows the parallel emergence of both Discontinuous (e.g. dark blue outline, trajectory shown) and Continuous (e.g. light blue outline, trajectory shown) modes under uniform conditions.

Cells have been segmented and tracked to highlight morphologies and trajectories. Note that two cells in the Continuous mode (yellow and lilac outlines) pass through the substrate region traversed by the Discontinuous mode cell (dark blue), yet these cells remain in the Continuous mode. This implies that these migration modes are not simply determined by (possible) local variations in, for example, ECM substrate (5 µg/ml fibronectin). Time in minutes shown.

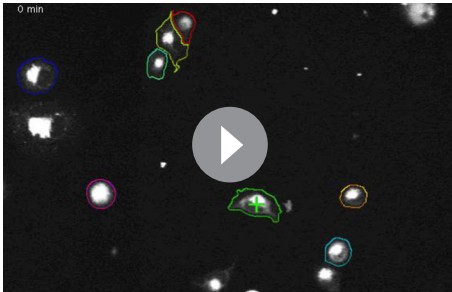

**Video 4.** Migration mode transition from Continuous to Discontinuous motility. Low-resolution imaging of H1299 P/L cells labeled with a membrane dye, during random migration on 5 µg/ml fibronectin. Cells have been segmented and tracked to highlight morphologies and trajectories. Note that the cell with light green outline (trajectory shown) transitions from Continuous to Discontinuous migration during the course of imaging. Time in minutes shown.

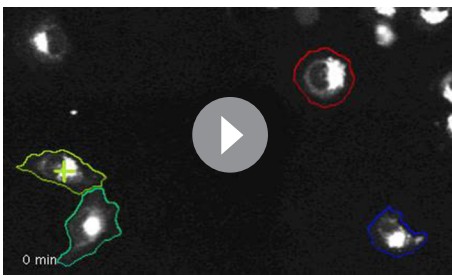

**Video 5.** Migration mode transition from Discontinuous to Continuous motility. Low-resolution imaging of H1299 P/L cells labeled with a membrane dye, during random migration on 5 µg/ml fibronectin. Cells have been segmented and tracked to highlight morphologies and trajectories. Note that the cell with light green outline (trajectory shown) transitions from Discontinuous to Continuous migration during the course of imaging. Time in minutes shown.

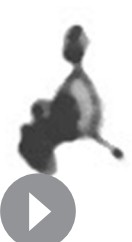

**Video 6.** BT549 cell migration in the Discontinuous mode. A single BT549 cell migrating in the Discontinuous mode. Cropped from a larger image montage. Cells labeled with a membrane dye during random migration on 2.5 µg/ml fibronectin. Image intensity scale inverted and brightness linearly adjusted for visualization (note: intensity data are not quantified). Time in minutes shown.

## Discontinuous mode membrane protrusions and retractions are less coordinated and temporally decoupled

To understand the temporal dynamics underlying differences in migration mode behavior, we assessed the cross-correlation between protrusion and retraction size signals over time. This revealed strong temporal coordination (high cross-correlation) between protrusion and retraction events during Continuous migration, with a single cross-correlation peak around the -1 time-lag. This indicated that protrusion consistently led retraction, but that these dynamics are tightly coupled in time (*Figure 2G*). This was true regardless of the time frame over which membrane dynamics were defined. In marked contrast, two cross-correlation peaks were apparent in the Discontinuous mode, with one peak stable around the -1 time-lag (tightly coupled dynamics), and a second peak temporally offset with positive lags (*Figure 2H*). This offset was dependent on the time frame used to define membrane dynamics, but clearly indicates a population of temporally decoupled membrane dynamics, wherein retraction leads protrusion events that are significantly delayed. This confirmed the visual impression of Discontinuous migration as a cyclical, stepwise process with asynchronous retraction (i.e. tail retraction stage) preceding protrusion (i.e. lateral protrusion stage). In contrast, the morphological stability observed during Continuous migration is a consequence of the tight coupling of protrusion and retraction, producing motility without large-scale, transient size/shape changes. Visual impressions also suggested the Discontinuous migration mode to be less ordered/coordinated than the Continuous mode. The significantly lower average (combining all time-lags) protrusion-retraction cross-correlation values during Discontinuous migration (*Figure 2I*) confirmed reduced coordination of membrane dynamics in this migration mode.

## Cell migration speed is determined by membrane retraction size rather than membrane protrusion size

To complete our analysis of how membrane dynamics differentially contribute to migration in each mode, we assessed the relationship between cell speed and the size of membrane protrusions (*Figure 2J*) or retractions (*Figure 2K*). Both modes exhibited similar dependencies. Remarkably, cell speed appears in each case to depend only slightly on the size of membrane protrusions, instead being largely determined by retraction size.

## The organizational states underlying Discontinuous and Continuous migration modes are quantitatively distinct

The data described above detail significant quantitative differences between both the cell motion and membrane dynamics comprising each migration mode behavior. Yet, how these divergent *behaviors* emerge from the *organization* of underlying macromolecular machineries remains unclear. We therefore extended our image analyses (see Materials and Methods and [*Lock et al., 2014*]) to derive a multivariate dataset of 55 organizational features (detailed in *Figure 3—figure supplement 1*) defining cellular-scale morphology as well as the state (e.g. size, number, density, morphology, localization) and dynamics (e.g. motion, stability, rates of area/density change) of critical macromolecular-scale machineries driving cell migration, namely, cell-matrix adhesion complexes (CMACs; demarcated by EGFP-paxillin) and F-actin (demarcated by RubyRed-LifeAct). This multiscale organizational data was measured per cell, per time-point, thereby complementing corresponding measures of migration behavior from the same individual cells (*Figure 3—figure supplement 1*). Such integrated data allowed exploration of the organizational states that give rise to particular cell migration behaviors, based on information embedded in natural cell heterogeneity (*Lock et al., 2014*; *Lock and Strömblad, 2010*).

To first understand how much overlap exists between the organizational states underpinning Discontinuous and Continuous migration modes, we performed supervised clustering (canonical vectors analysis, CVA) of all cell observations based on their organizational features only (*Figure 3A*). Strikingly, the two migration mode populations showed almost no overlap, indicating that their underlying patterns of organization can be quantitatively distinguished. We next visualized these organizational states via parallel coordinate mapping of 15 key organizational features (*Figure 3B*). This revealed specific instances of dissimilarity in feature values as well as an overview of the collective organizational signatures associated with each mode. To estimate the contribution of specific features to the divergence between modes, we ranked organizational features by the coefficient values assigned to them within the first canonical vector of the CVA (*Figure 3C*). We then compared feature value distributions between modes for a subset of these top-ranked features, finding highly significant differences in all cases (*Figure 3D–K*). These analyses help to define the spectrum of differences in organization that spontaneously arise in parallel with the emergence of Discontinuous and Continuous migration mode behaviors. For

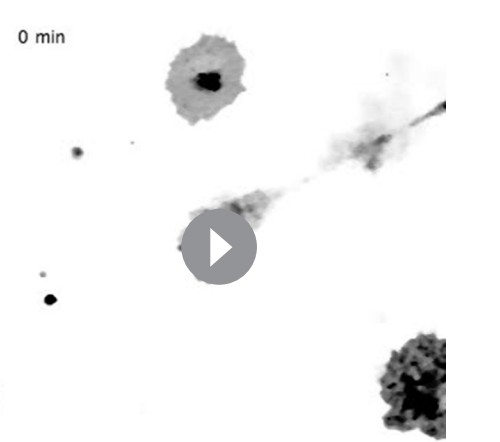

**Video 7.** BT549 cell migration in the Continuous mode. A single BT549 cell migrating in the Continuous mode (upper centre cell). Cropped from a larger image montage (note, montage stitching can cause observable intensity boundaries in image; however, these have no effect on interpretation). Cells labeled with a membrane dye during random migration on 2.5 µg/ml fibronectin. Image intensity scale inverted and brightness linearly adjusted for visualization (note: intensity data are not quantified). Time in minutes shown.

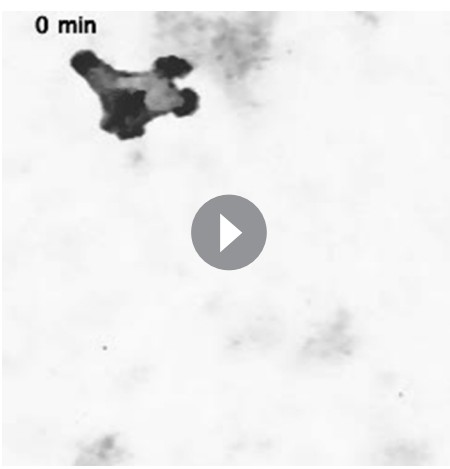

**Video 8.** MDA-MB-231 cell migration in the Discontinuous mode. A single MDA-MB-231 cell migrating in the Discontinuous mode. Cropped from a larger image montage. Cells labeled with a membrane dye during random migration on 2.5 µg/ml fibronectin. Image intensity scale inverted and brightness linearly adjusted for visualization (note: intensity data are not quantified). Time in minutes shown.

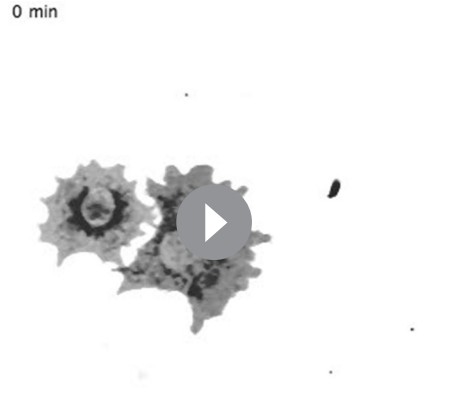

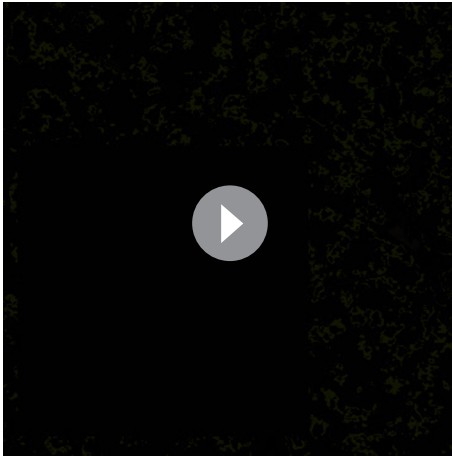

**Video 9.** MDA-MB-231 cell migration in the Continuous mode. MDA-MB-231 cells migrating in the Continuous mode. Cropped from a larger image montage. Cells labeled with a membrane dye during random migration on 2.5 µg/ml fibronectin. Image intensity scale inverted and brightness linearly adjusted for visualization (note: intensity data are not quantified). Time in minutes shown.

**Video 10.** Hep3 cell migration in the Discontinuous mode. Hep3 cells migrating in the Discontinuous mode. Cropped from a larger image montage (note, montage stitching can cause observable intensity boundaries in image; however, these have no effect on interpretation). Cells labeled with a membrane dye during random migration on 2.5 µg/ml fibronectin. Image intensity scale inverted and brightness linearly adjusted for visualization (note: intensity data is not quantified). Time in minutes shown.

instance, cells in the Discontinuous mode are: far smaller; less round (more protrusive); have fewer, smaller, less dense CMACs (based on EGFP-paxillin intensity) with less F-actin association (based on background-subtracted RubyRed-LifeAct intensity [*Li et al., 2010*]) and shorter lifetimes (less stable), resulting in a drastically decreased total adhesion area.

### Differences between inter-feature relationships in Discontinuous and Continuous migration modes appear highly selective

Having defined a spectrum of differences in key organizational features underlying migration modes, we next explored how the correlative relationships between these features might also vary. To this end, we mapped the complete network (including both organizational and behavioral features) of pairwise inter-feature Spearman's correlation coefficients (*rs*) for both Discontinuous (*Figure 4A*) and Continuous (*Figure 4B*) modes. Visual inspection of the heatmaps presented in 4A and 4B suggests that, while some differences in correlations are apparent, general patterns of correspondence are well preserved regardless of migration mode. This is emphasized by calculation of absolute differences in *rs* values between modes (*Figure 4C*), wherein the majority of relationships appear virtually unchanged. Indeed, only a very limited number of inter-feature relationships change by more than 0.4 *rs* (*Figure 4D*). This suggests that changes in inter-feature correlations are unexpectedly selective, given the magnitude of changes in the actual feature values themselves (*Figure 3*). To assess this, we compared the observed distribution of absolute differences in *rs* values to that which would be expected if changes arose randomly from the *rs* distributions depicted in *Figure 4A and 4B* (*Figure 4E*). This confirmed that observed changes are much smaller and less frequent than randomly expected, thus supporting the proposal that reconfiguration of correlative relationships is highly selective.

### Inversion of organizational feature correlations to cell speed between migration modes

We next analyzed how organizational features correlated to cell speed in each migration mode. As exemplified in *Figure 5A*, and more extensively in *Figure 5B*, we found that a proportion of such

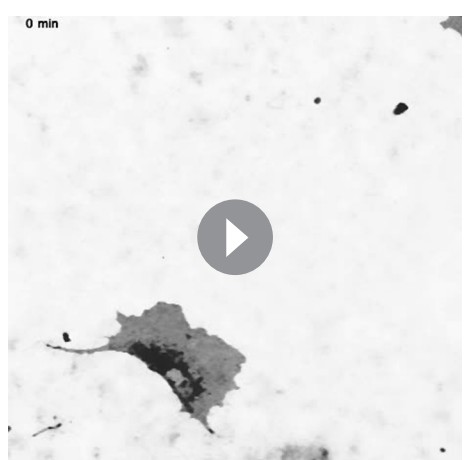

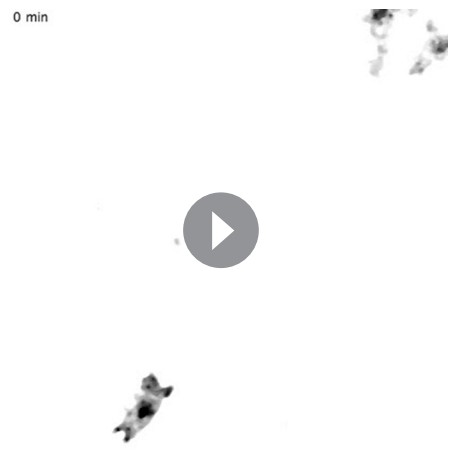

**Video 11.** HS578T cell migration in the Continuous mode. A single HS578T cell migrating in the Continuous mode. Cropped from a larger image montage. Cells labeled with a membrane dye during random migration on 2.5 μg/ml fibronectin. Image intensity scale inverted and brightness linearly adjusted for visualization (note: intensity data are not quantified). Time in minutes shown.

**Video 12.** H1299 P/L cell migration in the Discontinuous mode on laminin. H1299 P/L cells migrating in the Discontinuous mode. Cropped from a larger image montage. Cells labeled with a membrane dye during random migration on 2 μg/ml laminin. Image intensity scale inverted and brightness linearly adjusted for visualization (note: intensity data are not quantified). Time in minutes shown.

correlations were essentially equivalent between modes (e.g. Cell Area vs Cell Speed $rs$ = negative in both modes; Cell Compactness − Cell Speed $rs$ = positive in both modes). However, a large number of cell speed correlations were detectable in only one of the two modes (e.g. Median [CMAC Distance to Border] per Cell vs Cell Speed $rs$ = positive only in Continuous mode; Median [CMAC Area] per Cell vs Cell Speed $rs$ = positive only in Continuous mode). Even more salient, a substantial number of correlations to Cell Speed were inverted between modes (e.g. Median [CMAC Lifetime] per Cell vs Cell Speed $rs$ = negative in Discontinuous mode, positive in Continuous mode; Median [CMAC Average Trailing Edge Speed] per Cell vs Cell Speed $rs$ = positive in Discontinuous mode, negative in Continuous mode). Such inversions in organizational feature-to-Cell Speed relationships hint at profound differences in the mechanisms controlling each mode. We also note that, more broadly, most organizational feature-to-Cell Speed correlations were both stronger and more positive during migration in the Continuous mode (*Figure 5C*). This suggests that Continuous mode migration is more directly dependent on the state of CMAC and F-actin machineries, while alternative machineries may play more dominant roles during Discontinuous migration.

## Divergent migration strategies: different patterns of organizational state remodeling are coupled to cell speed variation in each migration mode

We have thus far detailed a variety of specific differences between Discontinuous and Continuous migration modes in terms of behavior (*Figure 1* and *Figure 2*), underlying organizational features (*Figure 3*) and individual inter-feature correlations (*Figure 4* and *Figure 5A–C*). We next explored how these differences might translate into systemic dissimilarities in the coupling of cell behavior (i.e. cell migration speed) and cell organization within each mode. To this end, we first grouped cell observations according to both migration mode and tertiles of cell speed (0−33% [slow]; 33−66% [medium]; 66−100% [fast]). We then performed supervised CVA clustering of these groups (*Figure 5D*). This enabled assessment of how cells in each mode cluster in the multivariate cell state space, but also of the trajectories defined through cell state space given variability in cell migration speed (as described previously, [*Lock et al., 2014*]) *within each mode*. As also observed in *Figure 3A* (based on similar but not equivalent clustering), we found that cells in Discontinuous or Continuous modes cluster separately. However, in a striking extension of this result, we also found that variability in migration speed within each mode corresponded to completely independent

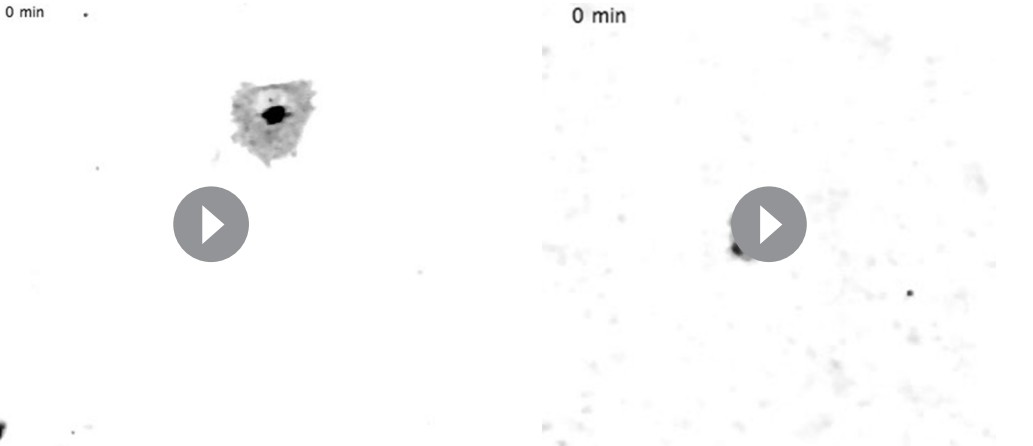

**Video 13.** H1299 P/L cell migration in the Continuous mode on laminin. A single H1299 P/L cell migrating in the Continuous mode. Cropped from a larger image montage. Cells labeled with a membrane dye during random migration on 50 μg/ml laminin. Image intensity scale inverted and brightness linearly adjusted for visualization (note: intensity data are not quantified). Time in minutes shown.

**Video 14.** H1299 P/L cell migration in the Discontinuous mode on collagen type 1. H1299 P/L cells migrating in the Discontinuous mode. Cropped from a larger image montage. Cells labeled with a membrane dye during random migration on 2 μg/ml collagen type 1. Image intensity scale inverted and brightness linearly adjusted for visualization (note: intensity data are not quantified). Time in minutes shown.

trajectories in cell state space. This means that different patterns of organizational remodeling are associated with varying cell speed within each mode. Thus, profound differences exist between modes in terms of the underlying strategies that produce and control cell migration speed.

## Cell-matrix adhesion and actomyosin contractility control the frequency equilibrium between mesenchymal migration modes

We have demonstrated how specific organizational feature values diverge in conjunction with the spontaneous emergence of Discontinuous and Continuous migration modes (*Figure 3*). Yet, it remains unclear whether these features are tightly coupled to migration mode identity, or only loosely correlated with these behaviors. We therefore assessed whether directed modulation of these features could drive corresponding and predictable changes in the frequency equilibrium between migration modes. To test for such bottom-up regulation, we applied a selection of targeted molecular perturbations for which we and others have established prior knowledge regarding their effects on the organizational features in question. These perturbations included: ECM ligand (FN) concentration modulation (*Gupton and Waterman-Storer, 2006*); talin 1 depletion (*Kiss et al., 2015*); and ROCK inhibition-mediated reduction of actomyosin contractility (*Lock et al., 2014*; *Kim and Wirtz, 2013*; *Hernández-Varas et al., 2015*) (*Figure 6A–C*). Images exemplifying perturbation effects on cell, CMAC and F-actin morphology are presented in *Figure 6—figure supplement 1*. Notably, these molecular perturbations collectively target four core regulatory mechanisms around the ECM – adhesion – F-actin axis, including integrin ligation; integrin activation/clustering; integrin-F-actin linkage; and actomyosin contractility (*Figure 6A*). Given the pivotal importance of these mechanisms for mesenchymal migration, the effects of the specific molecular perturbations applied herein may be somewhat predictive of a wide array of related regulatory mechanisms.

Importantly, we found that Continuous mesenchymal migration occurred upon high adhesion (high FN concentration) and with unperturbed actomyosin contractility, while Discontinuous migration occurred under conditions of low adhesion (low FN concentration) and inhibited contractility (ROCK-inhibitor). Talin knock-down promoted Discontinuous migration (*Figure 6C*). We then compared: i) spontaneous differences in feature values between Continuous and Discontinuous migration modes (displayed in *Figure 3D,E,G and I*); ii) perturbation-induced differences in the same



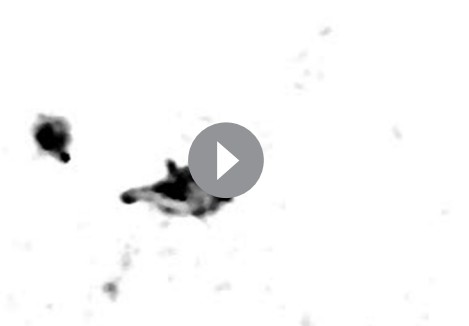

**Video 15.** H1299 P/L cell migration in the Continuous mode on collagen type 1. H1299 P/L cells migrating in both Discontinuous and Continuous modes. Cropped from a larger image montage. Cells labeled with a membrane dye during random migration on 5 µg/ml collagen type 1. Image intensity scale inverted and brightness linearly adjusted for visualization (note: intensity data are not quantified). Time in minutes shown.

**Video 16.** H1299 P/L cell migration in the Discontinuous mode on vitronectin. A single H1299 P/L cell migrating in the Discontinuous mode. Cropped from a larger image montage. Cells labeled with a membrane dye during random migration on 1 µg/ml vitronectin. Image intensity scale inverted and brightness linearly adjusted for visualization (note: intensity data are not quantified). Time in minutes shown.

features (*Figure 6B*); and iii) perturbation effects on migration mode frequencies (*Figure 6C*). From this matrix of results, we were able to determine whether a logical coherence was preserved over all these measures. For example, median CMAC Area was spontaneously higher in cells during Continuous migration (*Figure 3I*), and increasing ECM ligand concentration (FN) causes higher CMAC Area values (*Figure 6B*, upper left panel). It is therefore logically consistent (*if CMAC Area and migration mode behaviors are functionally coupled*) that raising ECM ligand concentration increases the frequency of the Continuous migration mode (*Figure 6C*, upper panel). In fact, although the applied perturbations produce differing effects on CMAC Area (increasing or decreasing), migration mode frequencies show logically consistent responses in each case. CMAC Area therefore appears likely to be closely coupled to migration mode identity, since there is no evidence that CMAC Area and migration mode identity could be uncoupled. Such coherence was also observed for CMAC Lifetime and Cell Compactness, thus similarly supporting their functional coupling to migration mode determination. In contrast, while the modulation of either ECM ligand concentration or talin 1 expression caused effects on Cell Area and CMAC Number that were consistent with their parallel effects on mode frequencies, ROCK inhibition caused contradictory results: CMAC number and Cell Area were both higher in cells in the Continuous mode (*Figure 3H*) and were increased following ROCK inhibition (*Figure 6B*, lower right panel), yet ROCK-inhibition caused a decreased frequency of the Continuous mode (*Figure 6C*, lower panel). These results imply that, although Cell Area and CMAC Number spontaneously correlate with migration mode identities under the original experimental condition, this correspondence can be uncoupled. These organizational features are therefore unlikely to be causal of, or caused by, migration mode identity determination.

Taken together, the matrix of results presented delineates organizational features (CMAC Area, CMAC Lifetime, Cell Compactness) that may be directly coupled with migration mode determination. More definitively, these results effectively exclude Cell Area and CMAC Number from being (linearly) causally linked to this process. Furthermore, these results demonstrate the potent regulatory influence of the specific molecular components fibronectin, talin 1 and ROCK, on migration mode determination. These perturbations also allows us to conclude more broadly that cell-matrix adhesion complexes and actomyosin contractility play key roles in shaping Discontinuous and Continuous mode frequencies.

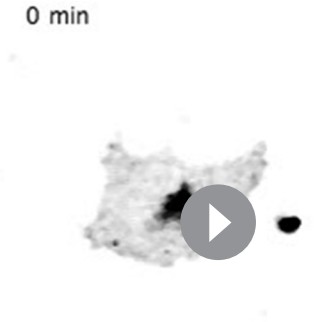

0 min

**Video 17.** H1299 P/L cell migration in the Continuous mode on vitronectin. A single H1299 P/L cell migrating in the Continuous mode. Cropped from a larger image montage. Cells labeled with a membrane dye during random migration on 1 µg/ml vitronectin. Image intensity scale inverted and brightness linearly adjusted for visualization (note: intensity data are not quantified). Time in minutes shown.

## Adaptive stretching or switching? Understanding the roles of mode remodeling and mode switching in adaptive responses

The molecular perturbations described above changed both cellular and macromolecular organization, as well as the frequency equilibrium between Discontinuous and Continuous migration modes. However, it remained unclear to what extent perturbation effects on organizational features may reflect: a) mode frequency changes (inter-modal 'adaptive switching'), as opposed to b) organizational remodeling of the modes themselves (intra-modal 'adaptive stretching'). To address the balance between these two adaptive responses, we leveraged data from the comparison of low (2.5 µg/ml) and high (10 µg/ml) FN concentrations displayed in *Figure 6*. Using only organizational features, we performed principal component analysis (PCA)-based unsupervised clustering of cell observations grouped by both FN concentration and migration mode (*Figure 7A*). Surprisingly, this revealed that the distances (in PCA-defined cell state space) between Discontinuous and Continuous migration modes were far greater than the distances induced within each mode by FN concentration variation. Comparing the heterogeneity of individual cells over time (*Figure 7B and C*, *Figure 7—figure supplement 1*) and the distances between migration mode centers of mass confirmed the substantial nature of average organizational differences between modes (equal to or greater than the maximal span of variation observed over 8 hr in the individual cells shown in *Figure 7B–C*). To more intuitively visualize how changing FN concentration shifts the distribution of cell organization within the PCA-defined cell state space (and between modes), we calculated the probability density distributions of cell observations in each mode, for each experimental condition (*Figure 7D–E*). This revealed that cell observations in each mode populate high probability-density 'valleys' that are at least partially separated by low probability-density 'ridges'. This visualization also emphasized that the major response to FN modulation was a substantial rebalancing of cell organizational states from one valley (migration mode) to the other. Collectively, these observations demonstrate that switching (between modes) dominates over stretching (remodeling within modes) as an adaptive response to FN concentration modulation. The conceptual relation between these two mechanisms of adaptation in cell state space is schematized in *Figure 7F*.

## Discussion

We here present an integrated analytical approach and associated tools, designed to interrogate both spontaneous and induced heterogeneity in single-cell mesenchymal (lamellipodial) migration. By deploying these tools, we quantitatively demonstrate that mesenchymal migration is composed of two distinct sub-modalities, each with specific behavioral, organizational, and regulatory characteristics. This marks an important advance that may enhance future investigations of the mesenchymal migration archetype. In particular, interpretations of natural and/or experimentally induced variability will likely become more coherent and precise if data are disaggregated according to Discontinuous or Continuous modes, rather than being unintentionally aggregated across cells with fundamentally different properties and dependencies. Recognition of these modes may therefore facilitate a more accurate understanding of the heterogeneity, adaptability, and regulation of mesenchymal migration as a whole.

Our initial classification of cells into Discontinuous and Continuous migration modes was based on observed behavioral differences. Crucially, the same distinctive behaviors emerged in parallel

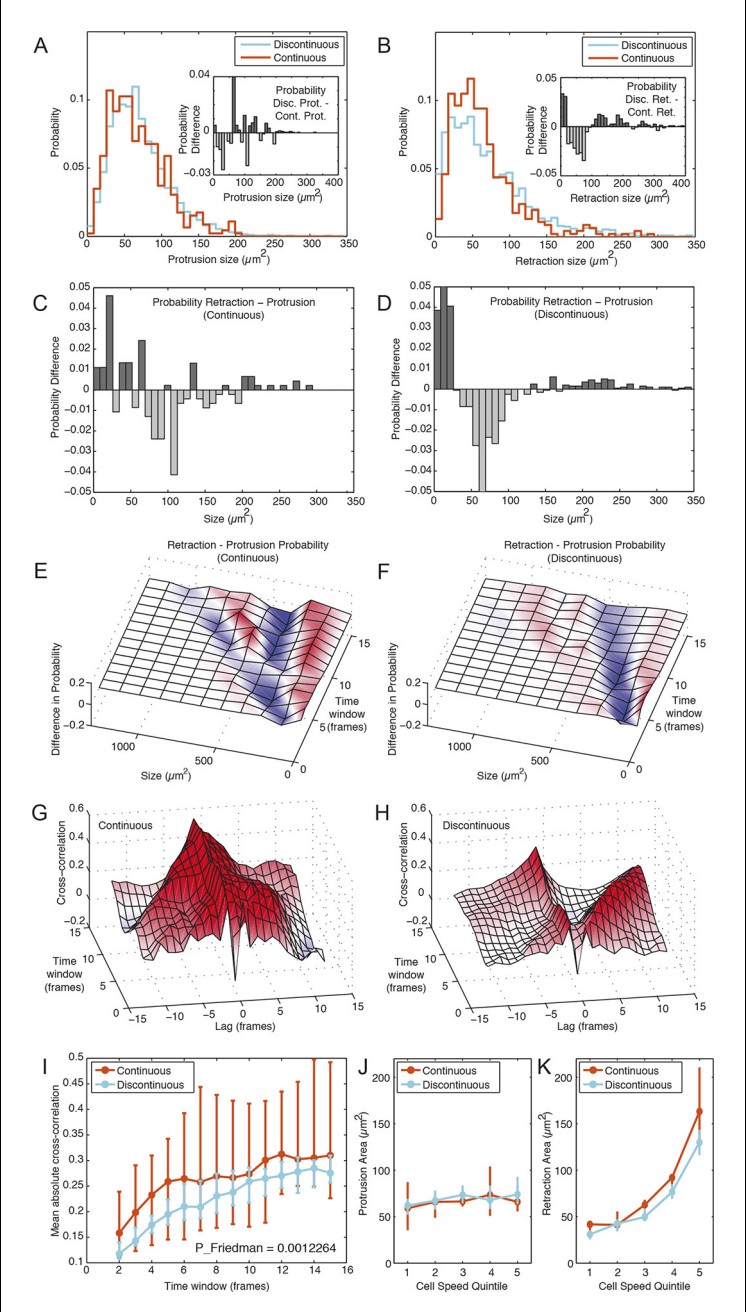

**Figure 2.** Membrane protrusion and retraction dynamics are temporally decoupled and less coordinated during Discontinuous migration. (**A**) Probability distribution of membrane protrusion sizes ($\mu m^2$) per 5-min interval during Discontinuous (blue) and Continuous (orange) migration. Subtraction of the Continuous from Discontinuous protrusion size probability distribution (inset) reveals no substantial or structured difference in these distributions (probability difference, Y axis). (**B**) Probability distribution of membrane retraction sizes ($\mu m^2$) per 5-min interval during Discontinuous (blue) or Continuous (orange) migration. Subtraction of the Continuous from Discontinuous retraction size probability distribution (inset) reveals that retraction sizes tend to be more extreme (frequently small or large, rarely moderate) in cells during Discontinuous migration (probability difference, Y axis). (**C**) Subtraction of the protrusion from retraction size probability distribution (5 min intervals only) during Continuous migration reveals little structure in probability differences. (**D**) Subtraction of the protrusion from retraction size probability distribution (5 min intervals only) during Discontinuous migration reveals that retractions in this mode are consistently more extreme in size (frequently small or large, rarely moderate) than protrusions. (**E**) Surface plotting of probability differences (retraction minus protrusion probability, per size) over various time windows (1 to 15 frames; 5 to 75 min) in cells during Continuous migration. Surface color-coding indicates where protrusions (blue)

*Figure 2 continued on next page*

*Figure 2 continued*

or retractions (red) are more common at a particular size. (**F**) Surface plotting of probability differences (retraction minus protrusion probability, per size) over various time windows (1 to 15 frames; 5 to 75 min) in cells during Discontinuous migration. (**G**) Signal cross-correlation was calculated between protrusion size and retraction size fluctuations per cell during Continuous migration. Mean cross-correlation values (Y axis, red = positive, blue = negative) plotted as a surface, per time lag (-12 to 12 frames, negative values indicate protrusion leads retraction, positive values indicate retraction leads protrusion), per time window (ranging from 1 to 15 frames; 5 to 75 min). (**H**) Signal cross-correlation was calculated between protrusion size and retraction size fluctuations per cell during Discontinuous migration. Mean cross-correlation values (Y axis, red = positive, blue = negative) plotted as a surface, per time lag (-12 to 12 frames, -ve values indicate protrusion leads retraction, +ve values indicate retraction leads protrusion), per time window (ranging from 1 to 15 time points; 5 to 75 min). (**I**) Mean (over all lags, per time window) absolute cross-correlation values for Continuous (orange) and Discontinuous (blue) migration, /- 95% confidence intervals, n = number of cells (see Materials and Methods). Statistical discernibility assessed by Friedman testing, p = 0.0012. (**J**) Cells in each migration mode were divided into quintiles (20% bins) based on instantaneous cell speed, and median protrusion areas /- 1.57 * interquartile range (IQR, 25% to 75%) / $\sqrt{n}$ (approximates 95% confidence interval of the median, n = number of cell observations, see Materials and Methods) were calculated per quintile. (**K**) Median retraction areas /- 1.57 * IQR/ / $\sqrt{n}$ (n = number of observations) were calculated per cell speed quintile.

within a variety of cell lines and under a spectrum of conditions, thus supporting the robustness and broad relevance of these behaviors. Yet, given the similarity of Continuous and Discontinuous migrations to previous descriptions of keratocyte-like (*Barnhart et al., 2015*; *Keren et al., 2008*) or fibroblast-like motility (*Abercrombie et al., 1977*; *Theisen et al., 2012*), respectively, the existence of these behaviors per se is not novel. Indeed, early observations even qualitatively suggest the co-existence of similar behaviors (*Lewis et al., 1982*). However, we now present quantitative evidence of the spontaneous, parallel emergence of two distinct yet inter-convertible mesenchymal migration modes, wherein divergent couplings exist between behavioral (speed) and organizational characteristics. We thereby provide new insights into the discontinuous structure of heterogeneity within the broad mesenchymal migration archetype. Indeed, by differentiating between progressive (intra-modal) and discrete (inter-modal) forms of variation, we gained the capacity to compare the roles that these two mechanisms play in adaptive responses to perturbation. Surprisingly, this revealed that adaptive switching between modes is the dominant response to perturbation (FN concentration modulation), with adaptive stretching (remodelling) within modes playing only a minor role.

The quantitative distinction between Discontinuous and Continuous migration modes is based on systemic differences detected via multivariate clustering of data following visual classification. Importantly, given that cells were classified based on perceived differences in their behavioral dynamics, we excluded behavioral features and focused only on organizational features (predominantly related to CMAC and F-actin status) when addressing the question of separation between the modes. As a result, at least four lines of evidence support the discrete nature of Discontinuous and Continuous migration modes.

First, supervised multivariate clustering (canonical vectors analysis, CVA) of mode-classified cell observations indicated near complete separation between Discontinuous and Continuous modes. The systemic organization underlying these migration modes is therefore, to a very large extent, quantitatively distinct.

Second, although separation between modes was not complete when cell observations were clustered in unsupervised PCA space, the topology of probability-density distributions was consistent with the existence of subpopulations in cell state space. Precisely, the high probability-density 'valleys' near the center of mass of each mode contrast with the low probability-density 'ridges' at the boundaries between modes. Importantly, these probability-density distributions are not predetermined outcomes of such PCA clustering, since PCA is unsupervised with respect to mode identity. In fact, these topologies are suggestive of a system comprised of two attractor states (*Huang and Ingber, 2000*), whose basins of attraction diverge on either side of a set of potentially unstable or unfavorable configurations. The low probability of these boundary configurations effectively divides mesenchymal migration into two independent sub-modalities. Such attractor states have been described in relation to, for example: gene expression during development and tumorigenesis

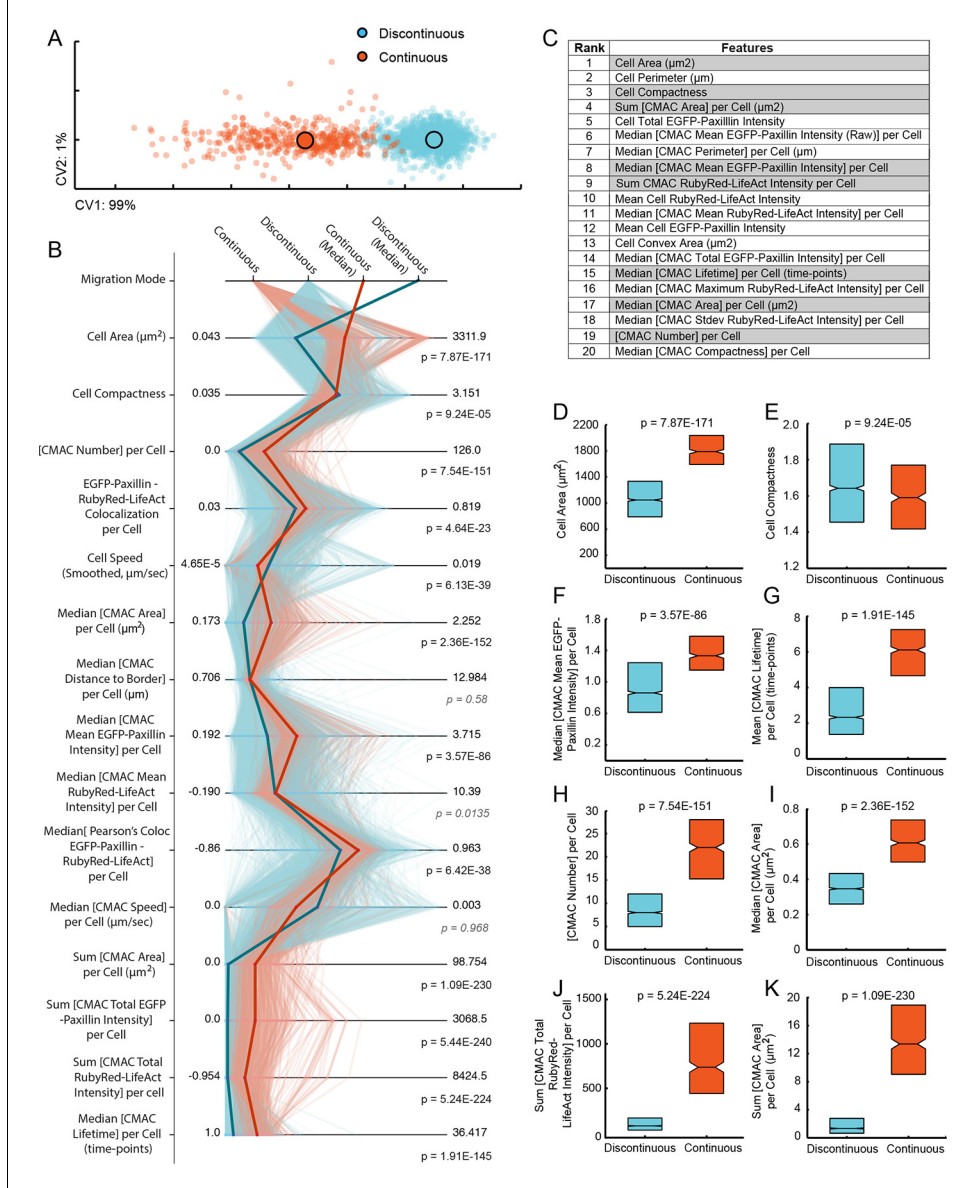

**Figure 3.** Discontinuous and Continuous migration modes reflect quantitatively distinct cell states with unique signatures of underlying organization. (**A**) Canonical vectors analysis (CVA)-based multivariate clustering (based on all 55 organizational features (see *Figure 3—figure supplement 1*), canonical vectors (CVs) 1 and 2 displayed, percentages indicate proportion of total variance per CV) of cell observations (see Materials and Methods) during Discontinuous (blue) or Continuous (orange) migration show modes to be quantitatively distinct. Large circles with black outlines indicate population centers of mass. (**B**) Parallel coordinate mapping of key organizational feature values (per cell observation) detail the multivariate signatures associated with cells during Discontinuous (light blue) or Continuous (light orange) migration. Points of difference between these multivariate signatures are emphasized by plotting of median values for Discontinuous (blue) and Continuous (orange) modes. Wilcoxon rank sum testing (per feature) assessed statistically discernable differences (p < 0.001), except where shown in gray italics. (**C**) A list of organizational features ranked by their contribution (coefficient values in canonical vector 1, which contains 99% of total variance) to the separation of migration modes in the canonical vectors space shown in (**A**). Gray backgrounds highlight features for which value distributions are compared between Discontinuous (blue) and Continuous (orange) modes in (**D-K**). (**D-K**) Features compared indicated on boxplot Y-axes. Boxplots show median values and inter-quartile ranges (IQR, 25% to 75%). Notches indicate median /- 1.57 * IQR/ / $\sqrt{n}$ (approximates 95% confidence interval of the median, n = number of cell observations, as in (**A**). P values reflect Wilcoxon rank sum testing as in (**B**).

The following figure supplement is available for figure 3:

**Figure supplement 1.** Definition of organizational and behavioral features.

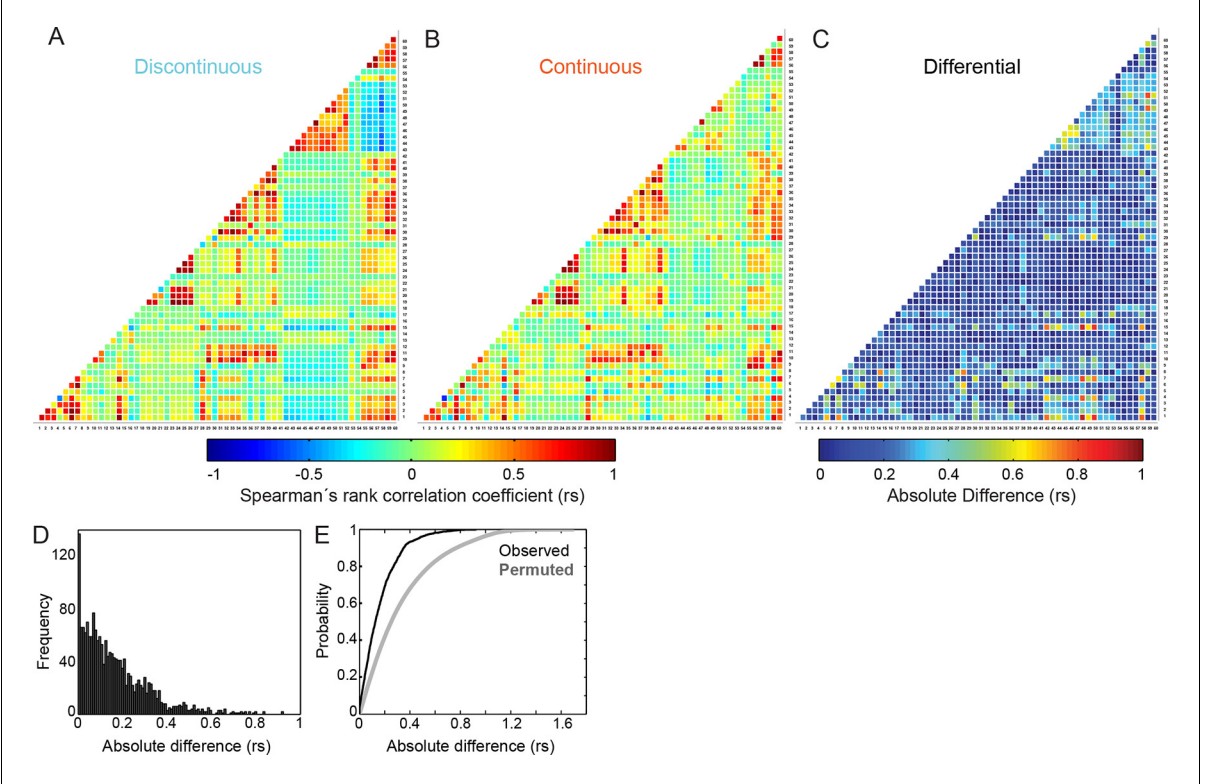

**Figure 4.** Differences in inter-feature relationships between Discontinuous and Continuous migration modes appear highly selective. (A, B) Heatmaps of Spearman's rank correlation coefficients (rs) for all pairwise combinations between 60 features (55 organizational and 5 behavioral), derived from cells during (A) Discontinuous or (B) Continuous migration. Individual correlation coefficient values are color-coded as depicted in the color bar (blue = negative; red = positive; green = near zero). Numbers on X and Y axes (1 to 60) correspond to the identities of features, as defined in Feature Number column of *Figure 3—figure supplement 1*. (C) A heatmap summarizes absolute differences in correlation coefficient values, per feature pair, between Discontinuous and Continuous migration modes. Difference values are color-coded as depicted in the color bar (blue = no difference; red = large difference). (D) A histogram shows the frequency distribution of absolute differences in correlation coefficient values between Continuous and Discontinuous migration modes. (E) A plot of cumulative distribution functions (CDFs) comparing the observed distribution of absolute differences in correlation coefficient values (between Discontinuous and Continuous modes as in [C] and [D], black line) and the differences in coefficient values obtained following randomized permutation of Spearman's correlation pairs (gray line). The permuted distribution (repeated 100 times for all relationships, all values included in CDF) shows the expected distribution of coefficient differences if inter-feature correlation changes occur randomly. Comparison of observed and permuted CDFs suggests that observed differences in inter-feature Spearman's correlation values are far more selective than randomly expected.

(*Huang et al., 2009*; *Huang, 2009*); cell signaling during motility (*Kim et al., 2015*); and even in the force-coupled dynamics of CMACs (*Hernández-Varas et al., 2015*). Moreover, the attractor state concept appears analogous to the concept of prespecification originally proposed by Friedl to explain the robust, recurrent and yet switchable characteristics of mesenchymal and amoeboid migration modes (*Friedl, 2004*). Interestingly, the limited remodelling within Discontinuous and Continuous modes in response to FN modulation (as opposed to mode switching) suggests that these modes may be relatively inflexible, providing additional support for their proposed status as prespecified attractor states.

Third, further indicating the discrete nature of Discontinuous and Continuous migration modes, CVA clustered cell observations grouped by both migration mode and cell speed were again largely distinct, supporting the validity of the original CVA clustering result. In addition, this approach also provided a fourth compelling line of evidence by revealing completely independent trajectories through cell state space defined by cells migrating at different speeds within each mode. Remarkably, this indicates that changes in cell speed within each mode were coupled to substantially different patterns of organizational remodelling. These systemic differences were composed in part by specific disparities in Spearman's correlations between individual organizational features and cell

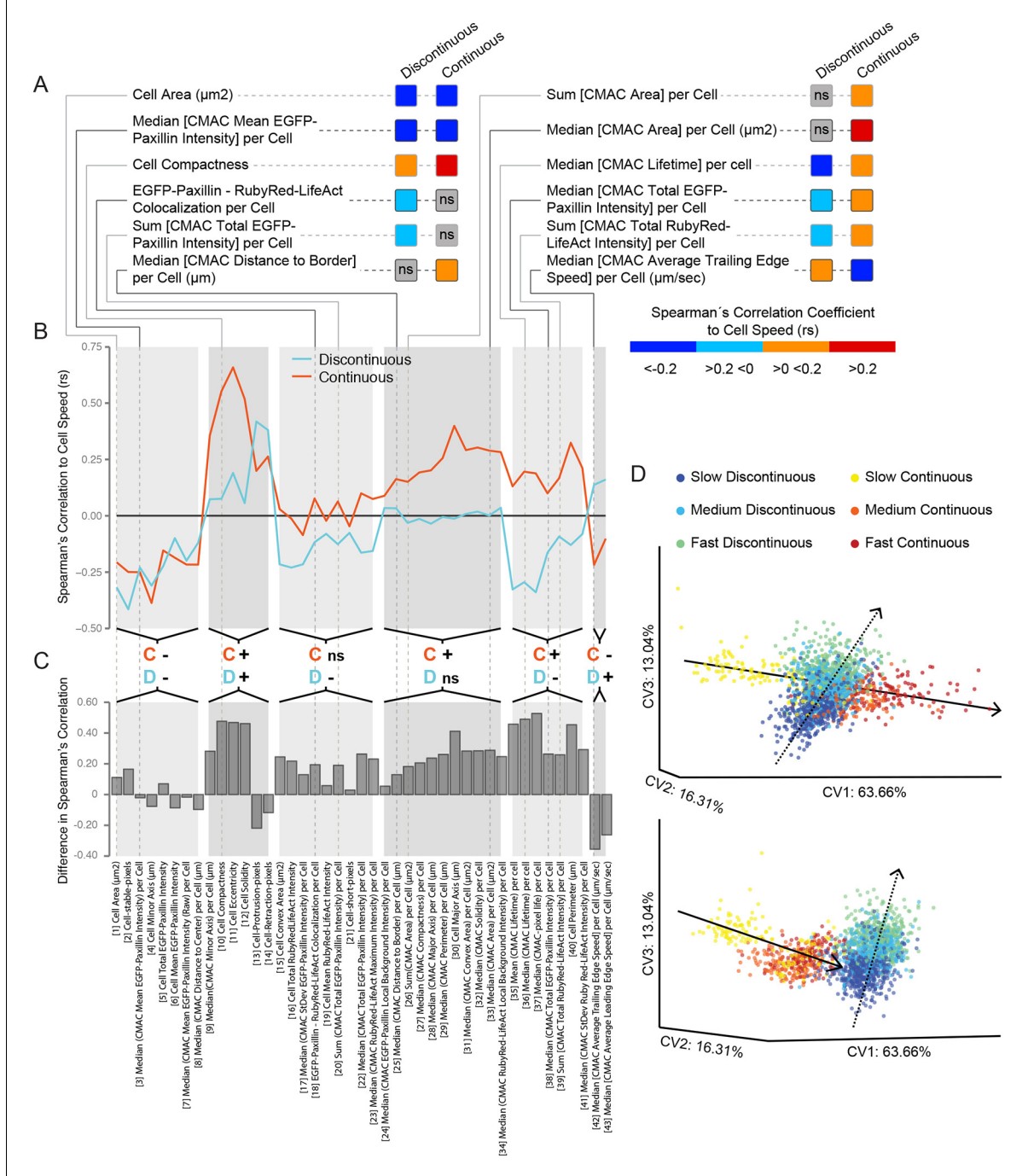

**Figure 5.** Cell speed variation is coupled to distinct patterns of remodeling in underlying features and inter-feature relationships within each migration mode. (**A**) A selection of Spearman's correlation coefficient values (rs) for organizational feature relationships to cell speed are depicted (color-coded as in color bar). Values reflect medians of populations values. Correlations are considered non-significant (ns) if zero is included in the range of the median +/−1.57 * inter-quartile range (IQR, 25% to 75%)√n (approximates 95% confidence interval of the median, n = number of cell observations, see 'Materials and methods'). This highlights correlations to cell speed that: are equivalent between modes (e.g. Cell Area, negative correlation in both modes); exist in only one mode (e.g. Median [cell-matrix adhesion complex (CMAC) Area] per Cell, ns in Discontinuous vs positive in Continuous); or are opposite between modes (e.g. Median [CMAC Lifetime] per Cell, negative in Discontinuous vs positive in Continuous). Examples in A are linked to an extensive analysis of cell speed-to-organizational feature correlations in B and C. (**B**) Parallel coordinate mapping of median cell speed-to-organizational feature correlations in Discontinuous (blue) and Continuous (orange) modes. Relationships are categorized by correlation value similarity or difference between modes, as indicated by the orange; C' (denotes Continuous) and blue 'D' (denotes Discontinuous) followed by '−'; (negative rs) or '+' (positive rs) or 'ns' (non-significant rs). (**C**) Bar graphs depict the magnitude of differences in median cell speed-to-organizational feature

*Figure 5 continued on next page*

*Figure 5 continued*

correlation values from the Discontinuous to the Continuous mode. Correlations tend to be stronger and more positive in cells during Continuous migration. (**D**) Canonical vectors analysis (CVA)-based multivariate clustering (using all 55 organizational features, canonical vectors [CV] 1, 2, and 3 displayed, percentages indicate proportion of total variance per CV) of cell observations during: slow Discontinuous (0–33.33% of Discontinuous migration speed values, blue); medium Discontinuous (33.34–66.66%, cyan); fast Discontinuous (66.67–100%, green); slow Continuous (0–33.33% of Continuous migration speed values, yellow); medium Continuous (33.34–66.66%, orange); or fast Continuous (66.67–100%, red) migration. Two orientations of the same three-dimensional clustering are depicted (upper and lower), revealing the separate clustering of each migration mode. Within each mode, progressive differences in cell speed correspond to similarly progressive variations in the position of observations within the multivariate organizational feature (or cell state) space. These speed-dependent trends in clustering define trajectories in the cell state space along which cells evolve as cell speed changes. Remarkably, these trajectories are completely distinct when comparing Discontinuous (dashed lines) and Continuous (solid lines) migration modes.

speed within each mode. While a variety of differences were observed, most striking were instances where correlations to cell speed were inverted between modes, such as in the case of adhesion stability (CMAC lifetime) – a critical parameter previously linked to causal regulation of cell speed (*Lock et al., 2014*). Interestingly, when comparing modes, the independent patterns of cell speed-coupled reorganization arose despite only subtle differences across the entire network of inter-feature Spearman's correlations. Indeed, these differences in correlation values were far more limited than would be expected at random, suggesting that a select few features and relationships may control migration mode identity.

Taking into account each of these lines of evidence, we conclude that Discontinuous and Continuous migration modes are representative of more than simply cells with different organizational feature values. We propose that these modes in fact represent divergent, yet co-existing strategies of mesenchymal migration.

Despite strongly embracing quantitative imaging-based research (*Lock and Strömblad, 2010*; *Le Dévédec et al., 2010*; *Masuzzo and Martens, 2015*), the cell migration field has seen remarkably little application of automated approaches to the classification of migrating cells. This is in contrast to the study of mitosis, where automated classification has already proven to be a powerful research tool (*Neumann et al., 2010*; *Schmitz et al., 2010*). It is likely that this reflects differences in the degree and structure of heterogeneity in these biological processes. For example, mitosis is a relatively stereotyped process - both morphologically and particularly in the terms of the temporal ordering of events. Both characteristics can increase the accuracy of automated classification approaches (*Held et al., 2010*). While cell migration is also more (e.g. Discontinuous mode) or less (e.g. Continuous mode) a globally time-ordered process, the underlying events appear far less stereotypical. Given these challenges, we have employed a blinded manual classification approach in this study. However, we note that the CVA clustering results – even excluding behavioral features (as herein) – provide evidence that automated classification of migration modes may be feasible in future. This may significantly enhance subsequent quantitative analyses of mode organization, behavior, and regulation.

Such automated classification may be in part based on our detailed spatiotemporal analysis of protrusive and retractive membrane dynamics within each mode. This revealed that the major differences between modes were: the size of membrane retraction events; the order and temporal coupling of protrusion and retraction events; and the coordination displayed between protrusive and retractive dynamics. Specifically, cells in the Discontinuous mode produced many very small and occasional very large retraction events, with these retractions tending to lead protrusion, reversing the order detected in Continuous cells. A large time delay also exists between retraction and subsequent protrusion dynamics in Discontinuous cells, and these dynamics are less coordinated (lower cross-correlation). Such substantial differences in membrane dynamics (especially the inversion of protrusion and retraction orders) further support the distinct nature of Discontinuous and Continuous migration modes. Furthermore, these analyses both confirmed and explained the visual impressions that originally differentiated Discontinuous from Continuous migration. In addition, given the strong correspondence between retraction (but not protrusion) size and cell speed, these analyses now add migration mode and cell speed to motility initiation as aspects of migration where membrane retraction appears to be the deterministic membrane process (*Barnhart et al., 2015*; *Kowalewski et al., 2015*; *Cramer, 2010*).

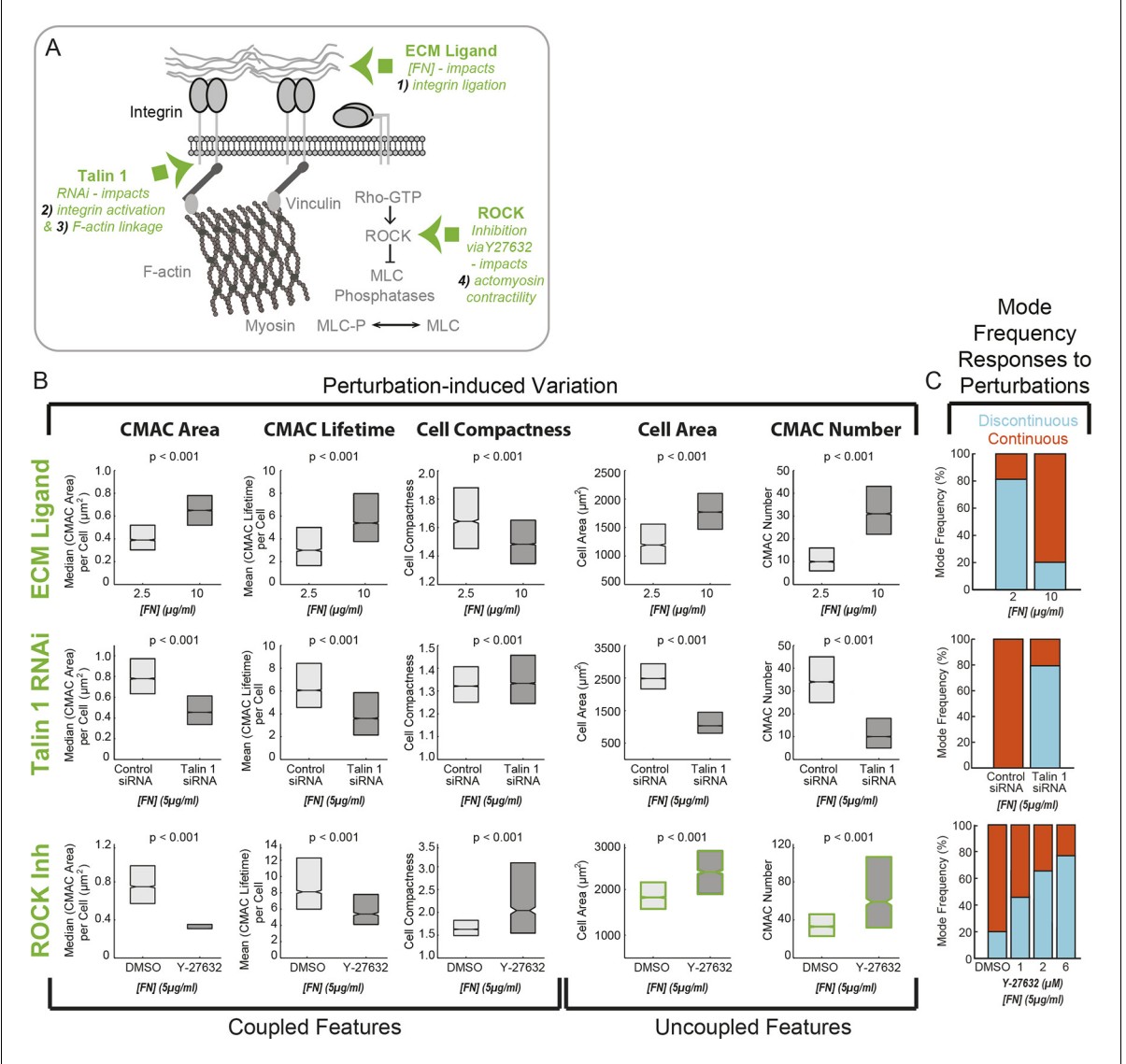

**Figure 6.** Cell-matrix adhesion and actomyosin contractility control the equilibrium between Continuous and Discontinuous migration modes. (**A**) Schematic summary of molecular mechanisms (green) associated with the ECM – adhesion – F-actin axis (gray) targeted here for perturbation to modulate implicated organizational features and, potentially, corresponding migration mode frequencies. Specifically, altering ECM ligand density (fibronectin concentration, [FN]) impacts mechanism 1) - integrin ligation and subsequent cell-matrix adhesion complex (CMAC) formation and maturation. RNAi-mediated knockdown of talin 1 limits both mechanism 2) - integrin activation and mechanism 3) - integrin-F-actin linkage, thereby also affecting CMAC formation, maturation and stability. Y-27632-mediated inhibition of ROCK disrupts mechanism 4) - actomyosin contractility, affecting F-actin dynamics and CMAC maturation. Images exemplifying the effects of each molecular perturbation are presented in *Figure 6—figure supplement 1*. (**B**) Boxplots summarizing the response of the same selection of organizational features to: ECM ligand modulation (2.5 μg/ml FN vs 10 μg/ml FN, upper row); talin 1 RNAi (control siRNA vs talin 1 siRNA, 5 μg/ml FN, middle row); or ROCK inhibition (DMSO vs 6 μM Y-27632, 5 μg/ml FN, lower row). All boxplots in B show median values per condition and inter-quartile ranges (IQR, 25% to 75%). Notches show the median $+/-1.57 * IQR/ / \sqrt{n}$ (approximates 95% confidence interval of the median, n = number of cell observations, see 'Materials and methods'). In each case, statistically discernable differences were assessed by Wilcoxon rank sum testing, with resulting p values <0.001. (**C**) Migration mode frequency responses to each perturbation are depicted (Discontinuous, blue; Continuous, orange). Additional conditions were included for ROCK inhibition (low panel), showing a progressive response to 1 μM, 2 μM, and 6 μM Y-27632 as compared to DMSO vehicle control. Note: in addition to depicting specific perturbation-dependent trends, the matrix of results presented in B and C is used, together with spontaneous feature variations depicted in *Figure 3D,E,G–I*, to logically parse organizational features that are consistently coupled, or just occasionally correlated, with migration mode identity.

The following figure supplement is available for figure 6:

*Figure 6 continued*

**Figure supplement 1.** Comparison of cell, adhesion complex and F-actin morphologies following perturbations targeting the ECM – adhesion – F-actin axis.

Having quantitatively established the discrete nature and behavioral characteristics of Discontinuous and Continuous migration modes, we sought to understand how their frequency balance is regulated. Importantly, to select candidate molecular regulators, we first considered the differences in organizational feature values that arose spontaneously between the modes. Then, given existing knowledge on how targeting of particular molecular components (FN concentration [*Gupton and Waterman-Storer, 2006*]; talin 1 expression [*Kiss et al., 2015*]; ROCK signaling [*Lock et al., 2014*; *Kim and Wirtz, 2013*; *Hernández-Varas et al., 2015*]) impact these same features, we designed a series of perturbations intended to drive specific changes in these features, while also monitoring the behavioral balance between modes. This demonstrated that each of these molecular factors plays a key role in shaping the equilibrium between modes. Furthermore, these molecular targets also regulate generic characteristics of the pivotal ECM – adhesion – F-actin axis, including: integrin ligation (FN concentration); integrin activation and clustering (talin expression); integrin – F-actin linkage (talin expression); and actomyosin contractility (ROCK signaling). Hence, the effects of these specific perturbations may also be illustrative of how generalized regulatory mechanisms impact the migration mode equilibrium.

Taking this perspective, we observe that while Continuous migration arises under conditions of high adhesion and full contractility, Discontinuous migration is preferred given low adhesion and inhibited contractility. It is noteworthy that both modes emerge under conditions that are distinct from those that favour amoeboid migration (i.e. low adhesion and high contractility). Therefore, the switch between Continuous and Discontinuous (mesenchymal) migration modes clearly contrasts to the previously identified conversion between mesenchymal and amoeboid migration. This emphasizes the multifaceted adaptive capacity of migrating cells, which likely expands the contexts within which efficient migration is possible. This plasticity is highlighted by the growing variety of distinct migration modalities now recognized (*Figure 7G*).

Finally, by viewing differences in organizational feature values (both spontaneous and perturbation-induced) and mode frequencies as a logical matrix, we could identify organizational features that are coupled to migration mode identity under all observed conditions, as well as features that can be explicitly uncoupled. This provides a first step in parsing correlated features from those with significant causal influence over migration mode identity. Specifically, it is notable that CMAC Area and CMAC Lifetime were previously indicated to causally influence cell migration speed (*Lock et al., 2014*; *Kim and Wirtz, 2013*), while Cell Compactness was causally downstream of cell speed (*Lock et al., 2014*). Thus, the evidence that these features may be functionally coupled to the determination of migration mode identity is consistent with them being integral to migratory regulation. Conversely, we previously found that Cell Area showed no causal relationship to cell speed (*Lock et al., 2014*), and this is again consistent with it being functionally uncoupled from the process of mode identity determination. Overall, three types of insight were revealed through these perturbation-based analyses, including: the regulatory roles of specific molecules (fibronectin, talin and ROCK); the putative influence of generic regulatory mechanisms; and the nature of links (coupled vs uncoupled) between migration mode identity and commonly measured macromolecular features. Nonetheless, much remains to be learned about the proximal and distal determinants of these migration modes.

In conclusion, we have presented and applied a quantitative, imaging-based analytic approach to explore the heterogeneity that naturally emerges within the mesenchymal (lamellipodial) archetype of cell migration. We thereby characterized two quantitatively distinct migration modes, here termed Continuous and Discontinuous migrations, that co-exist within the broad mesenchymal migration archetype. We compared these migration modes in terms of: their motion and behavioral dynamics; the specific and systemic organization of key machineries (CMACs and F-actin) underlying their motility, and; differences in how behavior (migration speed) and organization co-evolve within each mode. Through targeted molecular perturbations, we defined both specific molecular and general mechanisms of bottom-up control over migration mode determination, while also beginning to parse potentially functional (coupled) relationships (between features and modes) from those that are

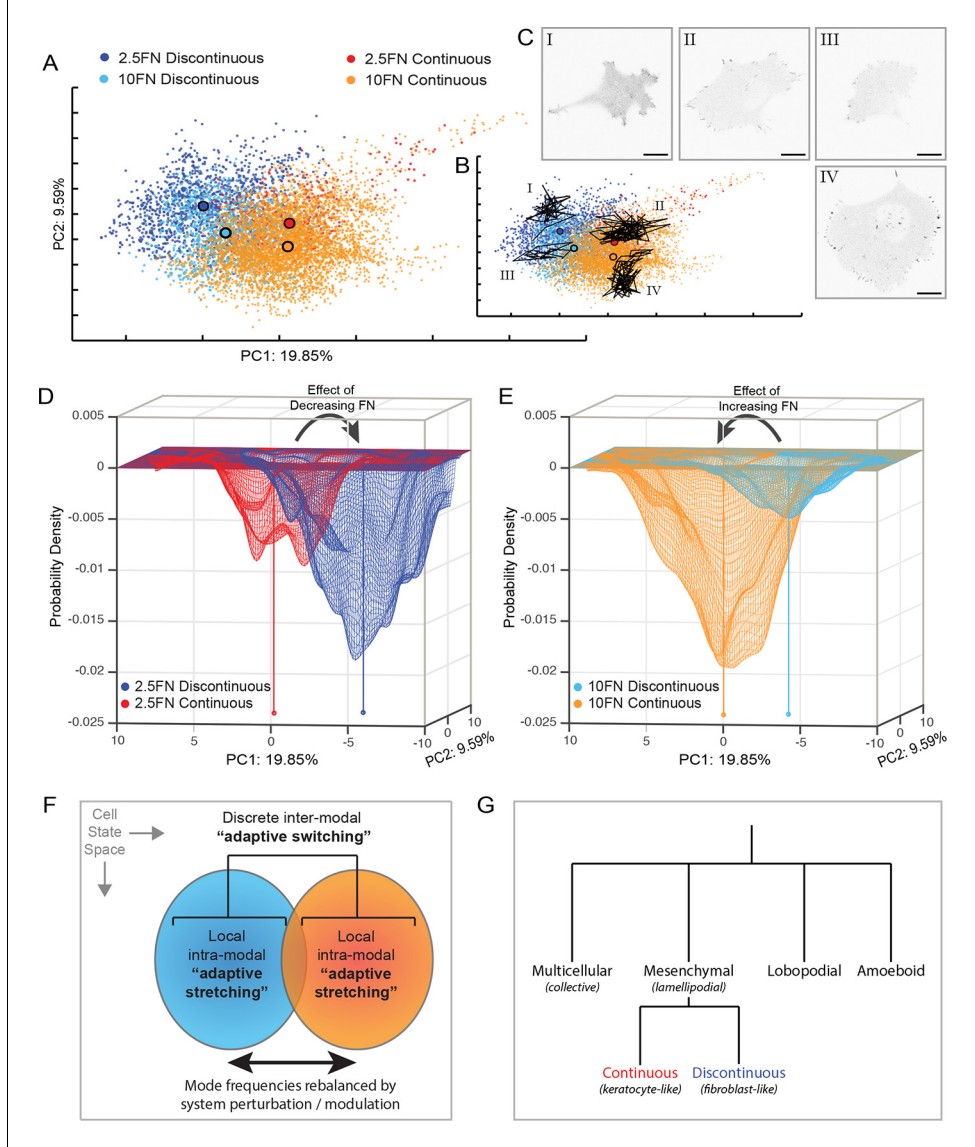

**Figure 7.** Adaptive switching between migration modes is the dominant perturbation response rather than intra-modal remodeling. (**A**) Principal component analysis (PCA)-based clustering of cell observations (based on all 55 organizational features), principal components (PC) 1 and 2 displayed, percentages indicate proportion of total variance per PC, color-coded by migration mode and experimental condition: Discontinuous mode on 2.5 μg/ml FN (dark blue); Discontinuous mode on 10 μg/ml FN (light blue); Continuous mode on 2.5 μg/ml FN (red); Continuous mode on 10 μg/ml FN (orange). Large black outlined circles indicate population centers of mass. Note that, despite FN variation exerting large effects on cell organization, differences between migration modes are much larger that differences between conditions within the same mode. (**B**) PCA clustering as in (**A**), overlaid by the heterogeneous trajectories over time of 4 cells (I–IV) within the PCA-based multivariate state space. Cells I–IV belong to the spatially corresponding cell populations defined in (**A**) and are relatively constrained within their respective regions of the cell state space. (**C**) Single time point confocal images of EGFP-paxillin from Cells I–IV from (**B**). Images are displayed with gray-scale inverted. Corresponding image sequence selections are presented in *Figure 7—figure supplement 1*. Scale bars = 20 μm. (**D**) A probability density map based on the coordinates in PCA space of cell observations from the 2.5 μg/ml FN condition in (**A**) (view orientation relative to A is rotated 180 degrees). Cells in the Discontinuous mode (dark blue) define a deep probability valley that partially overlaps with but is largely distinct from a shallow probability valley defined by cells in the Continuous mode (red). The sum of probabilities equals 1. (**E**) A probability density map based on the coordinates in PCA space of cell observations from the 10 μg/ml FN condition in (**A**) (view orientation relative to A is rotated 180 degrees). Cells in the Continuous mode (orange) define a deep probability valley that overlaps with but is partly distinct from a shallow probability valley defined by cells in the Discontinuous mode (light blue). The sum of probabilities equals 1. Centers of mass for each population in (**D**) and (**E**) are indicated by capped vertical lines of matching colors. Arrows in (**D**) and (**E**) signify how decreasing or increasing FN concentration, respectively, causes 'switching' from one probability valley (migration mode) to the other. (**F**) Schematic summary of the dual adaptive strategies employed by cells, with respect to mesenchymal migration. Modulation of intracellular or extracellular conditions can cause the remodeling of cellular and macromolecular organization locally within a given mode ('adaptive stretching'). However, the

*Figure 7 continued on next page*

*Figure 7 continued*

much larger and more frequent response to such perturbations is 'adaptive switching' between discrete migration modes, resulting in substantial rebalancing of Discontinuous and Continuous migration mode frequencies. (**G**) Dendogram (qualitative) indicating the major recognized archetypes of cell migration (top row) and the new modalities of Continuous and Discontinuous mesenchymal migration described herein (bottom row). Italicized names in brackets correspond to similar/analogous migration modes/terms.

The following figure supplement is available for figure 7:

**Figure supplement 1.** Confocal image sequences of cells during Discontinuous and Continuous mode migration on glass coated with low or high fibronectin concentrations.

correlated but non-functional (uncoupled). Finally, our systemic analysis of adaptive responses to perturbation has revealed how discrete mechanisms (inter-modal adaptive switching) dominate over progressive remodeling (intra-modal adaptive stretching). This study therefore emphasize the importance of distinguishing and comprehending Discontinuous and Continuous migration modes as a necessary precursor to understanding mesenchymal migration in its totality, while simultaneously providing tools and approaches that enable this endeavor.

## Materials and methods

### The cell adhesion and migration analysis toolbox

We have integrated a comprehensive and unique combination of analytical capabilities and made them freely available as a Matlab toolbox: 'The Cell Adhesion and Migration Analysis Toolbox' – along with the raw quantitative data underlying this study, sample image data to aid implementation, and explanatory documentation (see doi:10.5061/dryad.9jh6m). Briefly, this toolbox includes the following features utilized in this study: Data import and preprocessing based on images and extracted variables (see *Quantitative Image Analysis*, below); Smoothing of cell and adhesion trajectories; Membrane protrusion and retraction dynamics extraction; Migration mode classification; Mean square displacement (MSD) of cell trajectories; Variable selection for uni- and multivariate analyses; Univariate statitstical analysis; Variable correlation based heat maps; Canonical vector analysis (CVA); Principal component analysis (PCA); and two dimensional kernel density estimation. A more detailed description is available within the Matlab Toolbox.

### Cell culture and experimental conditions

All cell lines were acquired directly from the American Type Culture Collection (ATCC, Manassas, VA), and therefore were not further authenticated. The cell lines used herein are not members of the ATCC list of commonly misidentified cell lines. H1299 (human non-small cell lung carcinoma, ATCC; ATCC# CRL-5803, mycoplasma negative) cells stably expressing EGFP-Paxillin and RubyRed-LifeAct, termed H1299 P/L cells, were established and maintained in RPMI 1640 medium (Gibco – Thermo Fisher Scientific, Waltham, MA) containing 400 µg/ml geneticin (G-418 sulfate, Gibco) with 10% fetal bovine serum (Gibco) and 1 mM glutamine, as described previously (*Lock et al., 2014*; *Kiss et al., 2015*). BT549 (ductal breast carcinoma, ATCC; ATCC# HTB-122, mycoplasma negative) cells were maintained in RPMI 1640 medium (Gibco) containing 10% fetal bovine serum (Gibco) and 1 mM glutamine. MDA-MB-231 (adenocarcinoma, ATCC; ATCC# HTB-26, mycoplasma negative), Hep-3 (hepatocellular carcinoma, ATCC; ATCC# HB-8064, mycoplasma negative) and Hs578T (breast carcinoma, ATCC; ATCC# HTB-126, mycoplasma negative) cells were maintained in DMEM (Gibco) containing 10% fetal bovine serum (Gibco). All cells were incubated at 37°C in 5% $CO_2$. In preparation for imaging, 96-well optical glass-bottomed plates (Zell-Kontakt, Nörten-Hardenberg, Germany) were coated with ECM ligands, including either collagen type 1 (Life Technologies – Thermo Fisher Scientific), laminin (Sigma-Aldrich, St. Louis, MO), fibronectin or vitronectin (both purified from human plasma as described previously [*Smilenov et al., 1992*; *Yatohgo et al., 1988*]). Coating was performed for 2 hr at 37°C followed by blocking with 1% heat-denatured bovine serum albumin (Sigma-Aldrich) for 1 hr at 37°C. ECM ligand coating concentrations were 10 µg/ml except where otherwise indicated (i.e. where fibronectin concentration was varied). RNAi-depletion of talin 1 was performed 48 hr prior to imaging or immunoblotting (with anti-talin [8d4, 1:500, Sigma-Aldrich] and

anti-tubulin [DM1A, 1:2000, Fisher Scientific] antibodies as described previously [*Kiss et al., 2015*]) using the following oligonucleotide sequence: (5' -GAA GAU GGU UGG CGG CAUU- 3') (synthesized by GenePharma, Shanghai, P.R. China). A non-targeting oligonucleotide sequence was used as control: (5' -GCG CGC UUU GUA GGA UUCG- 3'). Transfection of $2 \times 10^4$ cells was performed in 24-well plates using 20 pmol of siRNA and 2 µl RNAiMAX (Invitrogen – Thermo Fisher Scientific) according to the manufacturers instructions. Inhibition of Rho kinase (ROCK) was performed starting 1 hr prior to imaging using the Y-27632 inhibitor (Sigma-Aldrich) at 1 µM, 2 µM, or 6 µM. Di-methyl sulfoxide (DMSO, Sigma-Aldrich) was used as control.

## Live-cell confocal imaging

Live-cell imaging was initiated 2 hr after replating of $2 \times 10^4$ cells per well into ECM-coated 96-well glass-bottomed plates (see coating details above). High-resolution, multiscale imaging (of CMACs, F-actin and cell migration) was performed on a Nikon A1R confocal microscope using a PlanApo VC 60X/1.4 NA oil-immersion objective (Nikon, Amsterdam, Netherlands). Images were acquired for 8–10 hr at 5 min intervals with a pixel resolution of 0.21 µm. Long-term (16 hr at 5-min intervals), low resolution imaging to monitor for inter-modal transitions was performed on a Nikon A1R confocal microscope using a Plan Apo γ 20X/0.75 NA air objective and resonance scanner, producing a pixel resolution of 0.82 µm. 3×3 or 4×4 image montages were acquired and stitched. Prior to imaging, cells were labeled at 1:5000 with a far-red membrane dye (Cell Mask Deep Red, Thermo Fisher Scientific) for 2 hr and throughout imaging. Normal culture medium minus fetal bovine serum was used during all imaging, and cells were maintained at 37°C in 5% $CO_2$.

## Quantitative image analysis, data parsing, and standardization

Multiscale image sequences (Nikon ND2 files) were analyzed using PAD software (Digital Cell Imaging Laboratories, Keerbergen, Belgium), as described previously (*Lock et al., 2014*). Briefly, cell boundaries were detected using the RubyRed-LifeAct signal. Cells touching the image border in any given time frame were excluded. Within each segmented cell, CMACs greater than 0.05 µm$^2$ in area were segmented based on EGFP-Paxillin signal. Segmented cells and CMACs were then tracked based on nearest neighbor analysis. Cell trajectories were smoothed using smoothing splines. CMACs were excluded where tracking did not contain at least three consecutive time points. CMACs present in the first and last frames of each image sequence were excluded from the calculation of CMAC Lifetime. All segmentation and tracking was manually inspected and iteratively optimized through parameter tuning. Quantitative features characterizing cell, CMAC, and CMAC-associated F-actin properties were then extracted (*Figure 3—figure supplement 1*). CMAC intensity values per channel were corrected by subtraction of mean intensity values in a 1 µm radius around the CMAC, excluding other segmented CMACs. CMAC intensity data were further standardized per experimental repeat relative to the median value of CMAC intensities in the size range between 0.15 and 0.2 µm$^2$ within the low FN no INH condition. Cell motion was characterized by instantaneous cell speed or via analysis of mean squared displacement (MSD). Mean square displacement (MSD) was calculated using a moving window as previously described (*Khorshidi et al., 2011*). The window size was set to 24 frames (2 hr). The first 12 time lags ($t_{lag}$), up to 1 hr, of the MSD were fitted to Furth's formula:

$$\mathrm{MSD} = 4M\left(t_{\mathrm{lag}} - t_p(1 - e^{-t_{\mathrm{lag}}/t_p})\right),$$

thus measuring the migration coefficient $M$ and the persistence time $t_p$ at each time point. The observations were divided into quintiles of migration coefficient and within each such quintile the mean persistence time for each trajectory was calculated. For visualization of parallel mode emergence (*Figure 1—figure supplement 1* and *Video 3*) and mode transitions (*Figure 1—figure supplement 2* and *Videos 4,5*), long-term low-resolution image data was segmented and tracked (following advanced denoising, power 15) using a custom general analysis pipeline in NIS-Elements (v4.30, Nikon).

Collectively, data sets for comparison of fibronectin concentration, talin 1 expression and ROCK inhibition had structures as follows: 2.5 µg/ml FN (3 experimental repeats, 34 cells, 2525 cell observations, 29651 CMAC observations) vs 10 µg/ml FN (19 experimental repeats, 118 cells, 6528 cell observations, 213406 CMAC observations); talin 1 siRNA (4 experimental repeats, 101 cells, 6263 cell observations, 81675 CMAC observations) vs control siRNA (4 experimental repeats, 51 cells,

3154 cell observations, 131679 CMAC observations); High-resolution, multiscale imaging of 6 µM Y27632 (ROCK inhibitor, 4 experimental repeats, 17 cells, 985 cell observations, 44647 CMAC observations) vs DMSO control (10 experimental repeats, 34 cells, 2666 cell observations, 86794 CMAC observations); long-term, low resolution imaging of 6µM Y27632 (4 experimental repeats, 73 cells, 4478 cell observations) vs 2 µM Y27632 (4 experimental repeats, 71 cells, 5198 cell observations) vs 1 µM Y27632 (4 experimental repeats, 69 cells, 4885 cell observations) vs DMSO control (4 experimental repeats, 83 cells, 5968 cell observations). Note that the term 'experimental repeats' refers to independent biological repeats, not technical replicates. Data sample sizes were selected based on prior experience from related multiscale analyses in several previous studies (*Lock et al., 2014*). Samples were collected and analyzed without iteration, that is, all relevant available data were used and sample sizes were not altered to increase or decrease statistical power.

## Migration mode classification

Cells were classified into Discontinuous or Continuous migration modes according to the behavioral criteria described in *Figure 1* based on visual inspection and optimization between two individuals, using a custom platform established in Matlab (vR2013b, The Mathworks, Natick, MA). Single cells were assessed at each time point within each image sequence (inter-mode transitions, though infrequent, were assessed). All data presented herein were assessed concurrently and were both computationally blinded and randomized with respect to experimental conditions. Dead cells and cells in sustained contact were classified as null and excluded from subsequent analyses.

## Quantification of membrane dynamics

Membrane protrusion and retraction dynamics were quantified as described previously (*Kowalewski et al., 2015*). Briefly, consecutive segmented images were compared to define protrusions (new pixels in cell segment compared to previous frame) and retractions (pixels lost from cell segment compared to previous frame). To explore dynamics over different time frames, images separated by different intervals were assessed using the same criteria, establishing time window samplings of between 5 min (1 frame) and 75 min (15 frames). Subtraction of protrusion and retraction size probability distributions was performed as described previously for other feature probability distributions (*Kiss et al., 2015*). Analysis of cross-correlation between protrusion and retraction areas over time was performed on a per cell sequence basis using the Matlab function xcov. Missing values were replaced by the mean value of protrusion/retraction area for that cell trajectory. Time lags of between −60 min (−12 frames) and 60 min (+12 frames) were assessed, with negative lags indicating that protrusion dynamics lead retraction dynamics, and positive lags indicating that retraction dynamics lead protrusion dynamics. To summarize the mean absolute cross-correlation per migration mode, cross-correlation values at all lags (−60 to 60 min; −12 to 12 frames) were averaged for each time window (5 min to 75 min; 1 to 15 frames), per cell, and then averaged per condition. Nienty-five percent confidence intervals were calculated per time window size. Friedman testing based on distinct cell numbers defined statistically discernable differences in mean protrusion-retraction cross-correlation between modes across the sampled time windows. All membrane dynamics analyses were performed in Matlab (vR2013b, The Mathworks).

## General statistical analyses and data visualization

Supervised multivariate clustering via canonical vectors analysis (CVA) was performed based on decompositions of the between-group and within-group covariance matrices following eigenvalue decomposition, as previously described (*Lock et al., 2014*). Feature selection via CVA was performed by ranking absolute feature weighting coefficients from the first canonical vector (CV1). Unsupervised multivariate clustering using principal component analysis (PCA) was performed using singular value decomposition of the normalized data matrix. Probability density surfaces were calculated using a Gaussian kernel density estimation of significantly smaller scale that the overall mode distributions, and hence does not dictate the 'valley' shape of these distributions – rather this is a reflection of the true value distribution. Spearman's rank correlation coefficients were calculated based on feature values per cell observation. Estimation of the expected distribution of random absolute value changes in Spearman's correlation between modes was performed by permuting the identity of correlation values in one mode while holding the other constant. This was repeated 100

times and the cumulative distribution function of all differences is displayed. Pairwise testing for differences in feature distributions (boxplots, parallel coordinates) was performed using the Wilcoxon rank sum test (equivalent to Mann-Whitney test) given all cell observations. Analyses and visualizations described above were performed in Matlab (The Mathworks). Parallel coordinates-based visualization of selected feature values in each migration mode was generated using Knime (v2.12.00, KNIME.com, Zurich, Switzerland). Parallel coordinates-based comparison of Spearman's correlation coefficients between Cell Speed and selected features, as well as visualization of coefficient differences between modes, were generated using R (v3.2.0, R Foundation for Statistical Computing) and RStudio (v0.98.945, RStudio, Boston, MA).

## Acknowledgements

HAS was supported by a scholarship by the Higher Education Commission of Pakistan. This work was supported by grants to SS from the EU-FP7–Systems Microscopy Network of Excellence (Grant No. HEALTH-F4-2010-258068), the Center for Innovative Medicine at KI, the Swedish Research Council (Grants No. 340-2012-6001 and 521-2012-3180), and the Swedish Cancer Society. Imaging occurred at the live cell-imaging facility and Nikon center of excellence at the Department of Biosciences and Nutrition at KI, supported by grants from the Knut and Alice Wallenberg Foundation, the Swedish Research Council, the Centre for Innovative Medicine, the Jonasson donation to the School of Technology and Health, and the Royal Institute of Technology, Stockholm, Sweden. The funders had no role in study design, data collection and analysis, decision to publish, or preparation of the manuscript.

## Additional information

### Funding

| Funder | Grant reference number | Author |
|---|---|---|
| European Commission | HEALTH-F4-2010-258068 | Staffan Strömblad |
| Vetenskapsrådet | 340-2012-6001 and 521-2012-3180 | Staffan Strömblad |
| Cancerfonden | | Staffan Strömblad |
| Center for Innovative Medicine at Karolinska Institutet | | Staffan Strömblad |
| Higher Education Commission, Pakistan | | Hamdah Shafqat-Abbasi |

The funders had no role in study design, data collection and interpretation, or the decision to submit the work for publication.

### Author contributions

HSA, Acquisition of data, Analysis and interpretation of data, Drafting or revising the article; JMK, AK, PHV, UB, MJM, SS, Analysis and interpretation of data, Drafting or revising the article; XG, Acquisition of data, Analysis and interpretation of data; JGL, Conception and design, Acquisition of data, Analysis and interpretation of data, Drafting or revising the article

### Author ORCIDs

Staffan Strömblad, http://orcid.org/0000-0002-1236-6339

## Additional files

### Major datasets

The following datasets were generated:

| Author(s) | Year | Dataset title | Dataset URL | Database, license, and accessibility information |
|---|---|---|---|---|
| Shafqat-Abbasi H, Kowalewski JM, Kiss A, Gong X, Hernandez-Varas P, Berge U, Jafari-Mamaghani M, Lock JG, Strömblad S | 2015 | Data from: An analysis toolbox to explore mesenchymal migration heterogeneity reveals adaptive switching between distinct modes | http://dx.doi.org/10.5061/dryad.9jh6m | Available at Dryad Digital Repository under a CC0 Public Domain Dedication |

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
