## [Decision Letter]

Thank you for submitting your work entitled "Heterogeneity in mesenchymal motility reflects adaptive switching between two distinct migration modes" for peer review at *eLife*. Your submission has been evaluated by Fiona Watt (Senior Editor) and three reviewers, one of whom, Johanna Ivaska, is a member of our Board of Reviewing Editors.

The reviewers have discussed the reviews with one another and the Reviewing editor has drafted this decision to help you prepare a revised submission.

Summary:

As you will see from the individual reviews included below, all the reviewers found that the paper is important and potentially suitable for publication in *eLife* as it describes fantastic analysis, with a level of detail and complexity that hasn't been attempted before. However, there was a consensus that the cell biological concepts put forward are somewhat overlapping with already published definitions regarding migration modes. Therefore, this article appears to be better suited to be published in *eLife* as a Tools and Resources article.

Essential revisions:

In case this would be something you would be happy with, please consider resubmitting the article formatted according to this category. This would mainly involve some re-writing to integrate your work better with the existing literature (please see the detailed comments below).

It would also benefit the cell motility field if you would be able to state explicitly how you think the discontinuous and continuous migration modes are related mechanistically to the other modes of carcinoma and fibroblast 2D and 3D migration. In addition, please consider possibly shortening the text in some parts (the Introduction for example is rather extensive), streamlining supplemental data and depositing the custom Matlab code appropriately.

Reviewer #1:

Much of the attention in the cell migration field has been in the mechanisms dictating the two main modes of cell migration, in particularly in 3D, namely amoeboid and mesenchymal migration modes. However, it is clear that not all mesenchymal cells migrate with the same morphological features. In this elegant, unbiased study Loch and co-workers describe two distinct modes on mesenchymal migration: continuous and discontinuous. The distinction is based on their high-though put microscopy and cutting edge image and statistical analyses of different migratory features. They define distinct features like cell adhesion number, localization and duration and their association with one of the two different migration modes. Furthermore, the dependency of the migration modes to perturbations of integrin activity, ECM ligand density and cell contractility are assessed in detail.

This is a carefully written manuscript putting forward an exciting new distinction of sub-modalities of mesenchymal migration.

Reviewer #2:

This manuscript characterises modes of migration on 2 dimensional surfaces which are relevant to mesenchymal cell motility. Using impressive multi-parametric analyses the authors determine interchangeable 'continuous' and 'discontinuous' migratory behaviours, which exhibit different requirements for specific features, for example continuous migration speed is closely correlated with cell matrix adhesion complex lifetime.

This study is elegantly designed and executed, and the characteristic properties of migrating mesenchymal cells are analysed in exquisite detail. Perturbation analysis is performed to analyse how specific properties might influence migratory mode. One major limitation of the study is the focus on a single cell line for the bulk of the analysis. Although other cells are more superficially investigated, in my opinion some extension is necessary to confirm the generalisability. For example, showing that manipulations can change the mode of migration in continuous-only (Hs578T) or discontinuous-only (Hep-3) cell lines would be particularly compelling.

My other concern with this manuscript in its current form is that it is not clear what has been determined that gives fresh insight and that it therefore may not constitute a significant advance of the highest novelty. This may be a harsh criticism, as it does constitute an excellent description of in depth analytical methodology.

Reviewer #3:

An important goal for the motility field is to understand how many different ways a single cell can migrate. This manuscript seeks to address this challenging question by using a combination of imaging and computational approaches to carefully characterize two types of mesenchymal cell migration defined here as continuous and discontinuous. Following a series of very careful comparisons of speed, persistence, protrusion dynamics, adhesion size, and F-actin intensity, for example, as well as comprehensive pairwise correlations between all of the parameters measured, they conclude that mesenchymal cells migrating on a two-dimensional tissue culture surface switch stochastically between distinct modes of migration (defined by an ensemble of cell-intrinsic parameters) in response to perturbations in cell-matrix adhesion and ROCK signaling, rather than using a continuous spectrum of migration mechanisms where each cell-intrinsic parameter can change independently. This conclusion is supported by the data presented and their discussion point that the presence of distinct migration mechanisms within population of cells need to be accounted for remains timely. It is difficult, however, to identify a clear conceptual advance resulting from this work or new information regarding the molecular mechanisms of cell movement.

Importantly, the authors have overlooked several recent papers where it was clearly shown that fibroblasts and other mesenchymal cells can use distinct migration mechanisms depending on the structure of the external environment, the degree of cell-matrix adhesion, as well as cell-intrinsic properties such as the level of RhoA/ROCK activity and actomyosin contractility (Liu et al, Cell, 2015, 160:659-672, Ruprecht et al, Cell, 2015, 160: 673-685, and Petrie et al, Science, 2014, 345:1062-1065). These publications each dilute the suggestion made by the authors here that it is unclear whether distinct sub-modalities of mesenchymal motility exist. Additionally, these papers and others (notably earlier publications from the labs of Peter Friedl, Erik Sahai, Ken Yamada, and Chris Marshall) clearly demonstrate the strong affect that matrix dimensionality has on the plasticity of cell migration mechanisms. How dimensionality could affect the migration of the cells used in this work is not addressed. Critically, the possibility that the continuous and discontinuous modes of migration characterized here only arise on an artificial tissue culture surfaces is not eliminated. It is also not clear why these cells need more than one mechanism to move.

Despite the lack of novelty in the conclusions of the work overall, the automated approach used to analyze cell motility is remarkable. This comprehensive and potentially unbiased approach has the clear potential to help identify new mechanisms of cell movement. For example, Figure 4C shows that the correlation between only a small subset of cell parameters changes significantly between the continuous and discontinuous modes. Identifying what those parameters are and if they actually caused the motility differences could be a way to leverage this data set to make new discoveries about the mechanisms regulating cell migration plasticity. Without this type of new mechanistic information I suggest the work might be more suited to be presented as a methods type paper.

[Editors' note: further revisions were requested prior to acceptance, as described below.]

Thank you for resubmitting your work entitled "Heterogeneity in mesenchymal motility reflects adaptive switching between two distinct migration modes" for further consideration at *eLife*. Your revised article has been favorably evaluated by Fiona Watt (Senior Editor) and the Reviewing Editor. The manuscript has been improved and modified to some extent but there are some remaining issues that need to be addressed before acceptance, as outlined below:

1) Novelty and main point of the paper: Although novelty is not highly important for the "Tool and Resources" category, the resubmitted title and Abstract both continue to emphasise the two modes of mesenchymal migration that are described and not on the automated and very complete analysis that they are able to perform. Very important is to highlight that the authors themselves accept in the Discussion that the existence of these two modes is not new and not even the co-existence of them. Please highlight more the quantitative information that you can extract and how this information can be correlated with responses to perturbation and the plasticity of cells to adapt to new conditions.

2) Repository of files: The manuscript mentions a "Cell adhesion and migration analysis toolbox" for Matlab with a detailed description of its functions and operation but it does not mention where is this placed. The rebuttal letter says that this "will be attached to the Data Dryad link" but still the location is not yet available. The scope of *eLife* and in particular of the "Tools and Resources" article type specifies that these articles should "highlight new experimental techniques, datasets, software tools and other resources" and also that "Tools and Resources articles should fully describe the biological material, data and methods so that prospective users have all the information needed to deploy them within their own work". At the moment the article does not fully describe the methods in the manuscript, maybe only in the detailed description coming with the Matlab toolbox but this is not accessible now. This needs to be revised.

---

## [Author Response]

Summary:

*As you will see from the individual reviews included below, all the reviewers found that the paper is important and potentially suitable for publication in eLife as it describes fantastic analysis, with a level of detail and complexity that hasn't been attempted before. However, there was a consensus that the cell biological concepts put forward are somewhat overlapping with already published definitions regarding migration modes.*

It is our belief that this issue may to some degree represent an important misunderstanding, stemming from the use of slightly ambiguous language on our part in the initial submission, for which we sincerely apologize.

It is absolutely critical to differentiate between a) the *mesenchymal modality of migration* – which is sometimes also referred to as “lamellipodial” or “lamellipodial-driven” migration, and b) the migration of *cells of mesenchymal origin* (which are known to migrate by various modes: mesenchymal; amoeboid; etc.). It is the *mode* of migration to which we refer in this paper, not to the migration of cells of mesenchymal origin. It is our view that while the existence of multiple modes of migration by cells of mesenchymal origin is not novel, the existence of quantitatively distinct modalities within the broader mesenchymal/lamellipodial migration modality is novel.

We recognize that the wording used in the original submission could lead to this unintended interpretation (leading to understandable questions about the novelty of findings), so we have made careful revisions in this language to clarify this subtle yet crucial issue. We hope that the novelty of our identification of quantitatively distinct, co-exiting modes of migration, within the mesenchymal (lamellipodial) modality of migration, will now be clearer.

We have chosen not to use “lamellipodial” or “lamellipodial-driven” migration as primary terms (though lamellipodial is now used in brackets as an ancillary term in Abstract and initial Introduction) because, as our data shows, it may be that the driving or determining process in mesenchymal migration is actually retraction of the cell rear. While lamellipodial protrusion and tail retraction are undoubtedly integrated process, we feel that the term lamellipodial migration may place too great an emphasis on this one facet of the migratory machinery/process.

Therefore, this article appears to be better suited to be published in eLife as a Tools and Resources article.

We feel that publication in the Tools and Resources format is an excellent opportunity and an appropriate place to present both the biological and methodological aspects of this work. We are grateful for the opportunity to do so.

*Essential revisions:*

*In case this would be something you would be happy with, please consider resubmitting the article formatted according to this category.*

We very much appreciate the opportunity to revise our manuscript into the Tools and Resources format, and have made extensive efforts to increase the emphasis, explanation and accessibility of analytical and statistical components of our research. At the same time, we have also worked on clarifying some of the language relating to our major questions and findings, to explain what we believe to be the legitimate and important novelty of our biological findings.

This would mainly involve some re-writing to integrate your work better with the existing literature (please see the detailed comments below).

We greatly appreciate the reviewers pointing out a number of important recent publications, which we had not previously referenced. We have ensured that these are now appropriately referenced, thereby enhancing the relevance and scope of the manuscript.

It is very important to note that, to our understanding, these and prior publications do not actually compromise the novelty of our findings, since they do not define quantitatively distinct, co-existing migration modes *within* the mesenchymal migration archetype (but rather between the mesenchymal mode and other migration modes). This distinction (between the mesenchymal migration modality, and migrating cells of mesenchymal origin) may have been a source of confusion, due to ambiguous language used in the initial submission. We have made every attempt to ensure that this issue is now absolutely clear.

It would also benefit the cell motility field if you would be able to state explicitly how you think the discontinuous and continuous migration modes are related mechanistically to the other modes of carcinoma and fibroblast 2D and 3D migration.

We have now stated explicitly how we think the Continuous (analogous to keratocyte-like migration) and Discontinuous (analogous to fibroblast-like migration) modes may relate to previously observed mesenchymal migration modes, while also highlighting that, though the modes may be analogous to previously described behaviors, we now actually demonstrate that they are quantitatively distinct (and not simply extremes in a phenotypic continuum), and that they co-exist within individual conditions (rather than only emerging independently given different cell, environmental or perturbation contexts). We have further clarified how we think these modes fit into the existing hierarchy of migration modalities by including a new summary figure panel (Figure 7).

In addition, please consider possibly shortening the text in some parts (the Introduction for example is rather extensive), streamlining supplemental data and depositing the custom Matlab code appropriately.

While we have included new (brief) text to clarify issues related to: the meaning of mesenchymal migration (modality, not cell type); the new suggested references; the research questions (clarifying the novel focus of the study); and the analytical research toolbox now described in this paper, we have still managed to shorten the overall Introduction through the removal of excess descriptions of the cell migration system. We agree with the reviewers that this will improve the readability and impact of the paper.

We made extensive efforts to combine (and compress) the supplementary data, i.e. movies, to improve the efficiency with which they could be accessed. However, we were unable to achieve this without exceeding the file sizes allowable by *eLife*. We are open to further suggestions (such as re-sizing the movies), however because this would involve changing the original data, we have not done this now.

According to the revising of this manuscript for the Tools and Resources category of *eLife* publications, we have made a major effort to package and present our “Cell Adhesion and Migration Analysis Toolbox”, i.e. the combination of statistical and analytical tools applied herein. Along with sample data sets (that will allow users to learn the approaches), this Matlab package will be attached to the Data Dryad link (along with the entire quantitative data-sets – already uploaded). We hope that this will provide complete transparency, aid reproducibility and reuse of these approaches, and assist interested researchers in applying the same or similar analytical approaches.

Reviewer #2:

[…] One major limitation of the study is the focus on a single cell line for the bulk of the analysis. Although other cells are more superficially investigated, in my opinion some extension is necessary to confirm the generalisability. For example, showing that manipulations can change the mode of migration in continuous-only (Hs578T) or discontinuous-only (Hep-3) cell lines would be particularly compelling.

We agree that this is an important suggestion, and it is our intention to follow this line of thinking in future, particularly by aiming to identify and broadly apply more selective signaling-based mechanisms to modulate mode identity. However, given the current scale of this paper, and the increased focus on the methodological aspect, we feel that increasing the data content of the paper may make it simply too heavy and unfocused.

*My other concern with this manuscript in its current form is that it is not clear what has been determined that gives fresh insight and that it therefore may not constitute a significant advance of the highest novelty. This may be a harsh criticism, as it does constitute an excellent description of in depth analytical methodology.*

We hope that our response to the question of novelty, associated with the second paragraph of the editors review summary, will to a large degree address this issue, even though we appreciate that Reviewer #2 still had a positive response to the manuscript. We believe that the changes in language made to clarify our reference to the mesenchymal modality of migration, rather than the migration of cells of mesenchymal origin, will make clearer the novel findings of this work.

Reviewer #3:

*[…] Importantly, the authors have overlooked several recent papers where it was clearly shown that fibroblasts and other mesenchymal cells can use distinct migration mechanisms depending on the structure of the external environment, the degree of cell-matrix adhesion, as well as cell-intrinsic properties such as the level of RhoA/ROCK activity and actomyosin contractility (Liu et al, Cell, 2015, 160:659-672, Ruprecht et al, Cell, 2015, 160: 673-685, and Petrie et al, Science, 2014, 345:1062-1065). These publications each dilute the suggestion made by the authors here that it is unclear whether distinct sub-modalities of mesenchymal motility exist.*

We hope that our response to the question of novelty, associated with the second paragraph of the editors review summary, will to a large degree address this issue, even though we appreciate that Reviewer #3 still had a positive response to the manuscript. We believe that the changes in language made to clarify our reference to the mesenchymal modality of migration, rather than the migration of cells of mesenchymal origin, will make clearer the novel findings of this work.

To clarify further, in relation to the suggested references, we note that in each case, the papers above discuss either a new, distinct mode of migration (e.g. lobopodial, Petrie et al.2014) or switching *between* the major archetypes/modalities of migration (e.g. mesenchymal, amoeboid [and blebbing]). They do not describe or quantify discrete switching between sub-modalities *within* these major migration archetypes, as we do.

Additionally, these papers and others (notably earlier publications from the labs of Peter Friedl, Erik Sahai, Ken Yamada, and Chris Marshall) clearly demonstrate the strong affect that matrix dimensionality has on the plasticity of cell migration mechanisms. How dimensionality could affect the migration of the cells used in this work is not addressed.

We agree that it is already clear that matrix dimensionality influences the balance between the major migration modalities, for example, the choice between collective, amoeboid and mesenchymal migration. However, since (sub-) modalities of the mesenchymal migration archetype had not previously been quantitatively defined, we believe it is still necessary to ask if the balance between these newly defined modes is also sensitive to matrix properties (e.g. FN density), adhesion regulation, and/or cellular contractility. Similarly, by investigating the regulation of the newly defined, quantitatively distinct modes, it is possible to ask (we believe for the first time in this context), whether responses to perturbations are principally mediated through either re-organization of the modes themselves (adaptive stretching), or through switching between modes (adaptive switching). This question and analysis are, we believe, unique, and highly relevant to both the modes we have defined, and possibly also those previously described.

Critically, the possibility that the continuous and discontinuous modes of migration characterized here only arise on an artificial tissue culture surfaces is not eliminated.

We are also eager to determine if these modes exist in more physiologically equivalent environments (e.g. simple 3D models and/or either in vivo or organotypic models). Nonetheless, we feel that since these more complicated experimental contexts are far less amenable to the detailed quantitative analyses performed herein, it is reasonable and valuable to present our findings and approach at this stage, to make the field aware both of the likelihood that distinct mesenchymal migration modes exist, as well as of analytical strategies to address this question.

It is also not clear why these cells need more than one mechanism to move.

We would agree that there may be no particular need for these cells to have more than one migration mechanism in the experimental context used. But, just as cells may differentiate along various trajectories – even in artificial conditions – we posit that these distinct modes reflect pre-specified cellular (attractor) states that have evolved to provide necessary plasticity to cells, given the diverse challenges they meet in vivo. While this remains to be confirmed, we believe that the robustness, reproducibility and recurrence (in various conditions and cells types) of these modes, supports the possibility that they have physiological relevance.

To address this point, we have also added in Discussion the notion that the capacity to switch between these modes provides the cells with enhanced adaptive capabilities. This concept is consistent with our finding that Continuous mesenchymal migration was favored (more frequent) upon strong adhesion and full contraction, while the Discontinuous mode dominated upon low adhesion and inhibited contraction. Thus, each of these conditions is distinct from those favoured by other known migration modes, e.g. amoebic migration (which occurs upon low adhesion and high contraction). The existence of the two mesenchymal modes may therefore expand the range of conditions within which the cells can migrate efficiently.

*Despite the lack of novelty in the conclusions of the work overall, the automated approach used to analyze cell motility is remarkable. This comprehensive and potentially unbiased approach has the clear potential to help identify new mechanisms of cell movement. For example,*
Figure 4*shows that the correlation between only a small subset of cell parameters changes significantly between the continuous and discontinuous modes. Identifying what those parameters are and if they actually caused the motility differences could be a way to leverage this data set to make new discoveries about the mechanisms regulating cell migration plasticity. Without this type of new mechanistic information I suggest the work might be more suited to be presented as a methods type paper.*

We appreciate the suggestion to emphasize the methodological aspects of this work, and have, as described above, taken extensive steps to emphasize, explain and share the analytical, statistical and conceptual aspects of this work.

[Editors' note: further revisions were requested prior to acceptance, as described below.]

1) Novelty and main point of the paper: Although novelty is not highly important for the "Tool and Resources" category, the resubmitted title and Abstract both continue to emphasise the two modes of mesenchymal migration that are described and not on the automated and very complete analysis that they are able to perform. Very important is to highlight that the authors themselves accept in the Discussion that the existence of these two modes is not new and not even the co-existence of them. Please highlight more the quantitative information that you can extract and how this information can be correlated with responses to perturbation and the plasticity of cells to adapt to new conditions.

In response to your suggestions, we have now made substantial revisions to the manuscript title and Abstract, and also re-ordered and/or modified text in the Introduction, Results, Discussion and Methods, to clearly reweight the emphasis of the paper from the biological to the methodological. In essence, we have attempted to emphasize the approach applied, and view the resulting biology through the lens of that methodology.

*2) Repository of files: The manuscript mentions a "Cell adhesion and migration analysis toolbox" for Matlab with a detailed description of its functions and operation but it does not mention where is this placed. The rebuttal letter says that this "will be attached to the Data Dryad link" but still the location is not yet available. The scope of eLife and in particular of the "Tools and Resources" article type specifies that these articles should "highlight new experimental techniques, datasets, software tools and other resources" and also that "Tools and Resources articles should fully describe the biological material, data and methods so that prospective users have all the information needed to deploy them within their own work". At the moment the article does not fully describe the methods in the manuscript, maybe only in the detailed description coming with the Matlab toolbox but this is not accessible now. This needs to be revised.*

In addition, we have inserted the DOI link to the Matlab toolbox, explanatory files, sample image data and complete raw quantitative data – hosted by Data Dryad. I apologize for the failure to include this link in the manuscript previously, this resulted from some confusion about the publication process.